# ON OPTIMALITY CONDITIONS FOR AUTO-ENCODER SIGNAL RECOVERY

## ABSTRACT

Auto-Encoders are unsupervised models that aim to learn patterns from observed data by minimizing a reconstruction cost. The useful representations learned are often found to be sparse and distributed. On the other hand, compressed sensing and sparse coding assume a data generating process, where the observed data is generated from some true latent signal source, and try to recover the corresponding signal from measurements. Looking at auto-encoders from this *signal recovery perspective* enables us to have a more coherent view of these techniques. In this paper, in particular, we show that the *true* hidden representation can be approximately recovered if the weight matrices are highly incoherent with unit $\ell^2$ row length and the bias vectors takes the value (approximately) equal to the negative of the data mean. The recovery also becomes more and more accurate as the sparsity in hidden signals increases. Additionally, we empirically also demonstrate that auto-encoders are capable of recovering the data generating dictionary when only data samples are given.

## 1 INTRODUCTION

Recovering hidden signal from measurement vectors (observations) is a long studied problem in compressed sensing and sparse coding with a lot of successful applications. On the other hand, auto-encoders (AEs) (Bourlard and Kamp, 1988) are useful for unsupervised representation learning for uncovering patterns in data. AEs focus on learning a mapping $\mathbf{x} \mapsto \mathbf{h} \mapsto \hat{\mathbf{x}}$, where the reconstructed vector $\hat{\mathbf{x}}$ is desired to be as close to $\mathbf{x}$ as possible for the entire data distribution. What we show in this paper is that if we consider $\mathbf{x}$ is actually *generated* from some true sparse signal $\mathbf{h}$ by some process (see section 3), then switching our perspective on AE to analyze $\mathbf{h} \mapsto \mathbf{x} \mapsto \hat{\mathbf{h}}$ shows that AE is capable of *recovering the true* signal that generated the data and yields useful insights into the *optimality* of model parameters of auto-encoders in terms of *signal recovery*. In other words, this perspective lets us look at AEs from a signal recovery point of view where forward propagating $\mathbf{x}$ recovers the true signal $\mathbf{h}$. We analyze the conditions under which the encoder part of an AE recovers the true $\mathbf{h}$ from $\mathbf{x}$, while the decoder part acts as the data generating process. Our main result shows that the true sparse signal $\mathbf{h}$ (with mild distribution assumptions) can be approximately recovered by the encoder of an AE with high probability under certain conditions on the weight matrix and bias vectors. Additionally, we empirically show that in a practical setting when only data is observed, optimizing the AE objective leads to the recovery of both the data generating dictionary $\mathbf{W}$ and the true sparse signal $\mathbf{h}$, which together is not well studied in the auto-encoder framework, to the best of our knowledge.

## 2 SPARSE SIGNAL RECOVERY PERSPECTIVE

While it is known both empirically and theoretically, that useful features learned by AEs are usually sparse (Memisevic et al., 2014; Nair and Hinton, 2010; Arpit et al., 2016). An important question that has not been answered yet is whether AEs are capable of recovering sparse signals, in general. This is an important question for Sparse Coding, which entails recovering the sparsest $\mathbf{h}$ that approximately satisfies $\mathbf{x} = \mathbf{W}^T \mathbf{h}$, for any given data vector $\mathbf{x}$ and overcomplete weight matrix $\mathbf{W}$. However, since this problem is NP complete (Amaldi and Kann, 1998), it is usually relaxed to solving an expensive optimization problem (Candes et al., 2006; Candes and Tao, 2006),

$$\arg\min_{\mathbf{h}} \|\mathbf{x} - \mathbf{W}^T \mathbf{h}\|^2 + \lambda \|\mathbf{h}\|_1 \tag{1}$$

where $\mathbf{W} \in \mathbb{R}^{m \times n}$ is a fixed overcomplete ($m > n$) dictionary, $\lambda$ is the regularization coefficient, $\mathbf{x} \in \mathbb{R}^n$ is the data and $\mathbf{h} \in \mathbb{R}^m$ is the signal to recover. For this special case, Makhzani and Frey (2013) analyzed the condition under which linear AEs can recover the *support* of the hidden signal. The general AE objective, on the other hand, minimizes the expected reconstruction cost

$$\mathcal{J}_{AE} = \min_{\mathbf{W}, \mathbf{b}_e, \mathbf{b}_d} \mathbb{E}_{\mathbf{x}} \left[ \mathcal{L}(\mathbf{x}, s_d \left( \mathbf{W}^T s_e(\mathbf{W}\mathbf{x} + \mathbf{b}_e) + \mathbf{b}_d \right)) \right] \tag{2}$$

for some reconstruction cost $\mathcal{L}$, encoding and decoding activation function $s_e(.)$ and $s_d(.)$, and bias vectors $\mathbf{b}_e$ and $\mathbf{b}_d$. Notice however, in the case of auto-encoders, the activation functions can be non-linear in general, in contrast to the sparse coding objective. In this paper we consider linear decoding activation $s_d$ because it is a more general case [1]. In addition, in case of AEs we do not have a separate parameter $\mathbf{h}$ for the hidden representation corresponding to every data sample $\mathbf{x}$ individually. Instead, the hidden representation for every sample is a parametric function of the sample itself. This is an important distinction between the optimization in eq. 1 and our problem – the identity of $\mathbf{h}$ in eq. 1 is only well defined in the presence of $\ell^1$ regularization due to the overcompleteness of the dictionary. However, in our problem, we assume a true signal $\mathbf{h}$ generates the observed data $\mathbf{x}$ as $\mathbf{x} = \mathbf{W}^T\mathbf{h} + \mathbf{b}_d$, where the dictionary $\mathbf{W}$ and bias vector $\mathbf{b}_d$ are fixed. Hence, what we mean by *recovery* of sparse signals in an AE framework is that if we generate data using the above generation process, then *can the estimate* $\hat{\mathbf{h}} = s_e(\mathbf{W}\mathbf{x} + \mathbf{b}_e)$ *indeed recover the true* $\mathbf{h}$ *for some activation functions* $s_e(.)$, *and bias vector* $\mathbf{b_e}$? *And if so, what properties of* $\mathbf{W}, \mathbf{b}_e, s_e(.)$ *and* $\mathbf{h}$ *lead to good recovery?* However, when given an $\mathbf{x}$ and the true overcomplete $\mathbf{W}$, the solution $\mathbf{h}$ to $\mathbf{x} = \mathbf{W}^T\mathbf{h}$ is not unique. Then the question arises about the possibility of recovering such an $\mathbf{h}$. However, as we show, recovery using the AE mechanism is strongest when the signal $\mathbf{h}$ is the sparsest possible one, which from compressed sensing theory, guarantees uniqueness of $\mathbf{h}$ if $\mathbf{W}$ is sufficiently incoherent [2].

## 3 DATA GENERATION PROCESS

We consider the following data generation process:

$$\mathbf{x} = \mathbf{W}^T\mathbf{h} + \mathbf{b}_d + \mathbf{e} \tag{3}$$

where $\mathbf{x} \in \mathbb{R}^n$ is the observed data, $\mathbf{b}_d \in \mathbb{R}^n$ is a bias vector, $\mathbf{e} \in \mathbb{R}^n$ is a noise vector, $\mathbf{W} \in \mathbb{R}^{m \times n}$ is the weight matrix and $\mathbf{h} \in \mathbb{R}^m$ is the true hidden representation (signal) that we want to recover. Throughout our analysis, we assume that the signal $\mathbf{h}$ belongs to the following class of distribution,

**Assumption 1.** *Bounded Independent Non-negative Sparse (BINS): Every hidden unit* $h_j$ *is an independent random variable with the following density function:*

$$f(h_j) = \begin{cases} (1 - p_j)\delta_0(h_j) & if & h_j = 0 \\ p_j f_c(h_j) & if & h_j \in (0, l_{\max_j}] \end{cases} \tag{4}$$

*where* $f_c(.)$ *can be any arbitrary normalized distribution bounded in the interval* $(0, l_{\max_j}]$, *mean* $\mu_{h_j}$, *and* $\delta_0(.)$ *is the Dirac Delta function at zero. As a short hand, we say that* $h_j$ *follows the distribution BINS($\mathbf{p}, f_c, \mu_{\mathbf{h}}, l_{\max}$). Notice that* $\mathbb{E}_{h_j}[h_j] = p_j\mu_{h_j}$.

The above distribution assumption fits naturally with sparse coding, when the intended signal is non-negative sparse. From the AE perspective, it is also justified based on the following observation. In neural networks with ReLU activations (*i.e.* $f_c(x) = \max(0, x)$), hidden unit pre-activations have a Gaussian like symmetric distribution (Hyvärinen and Oja, 2000; Ioffe and Szegedy, 2015). If we assume these distributions are mean centered[3], then the hidden units' distribution after ReLU has a large mass at 0 while the rest of the mass concentrates in $(0, l_{\max}]$ for some finite positive $l_{\max}$, because the pre-activations concentrate symmetrically around zero. As we show in the next section, ReLU is indeed capable of recovering such signals. On a side note, the distribution from assumption 1 can take shapes similar to that of Exponential or Rectified Gaussian distribution[4] (which are generally used for modeling biological neurons) but is simpler to analyze. This is because we allow $f_c(.)$ to be

---

[1] Linear decoding activation covers numerical range for commonly used non-linear decoding activations

[2] Coherence is defined as $\max_{\mathbf{W}_i, \mathbf{W}_j, i \neq j} \frac{|\mathbf{W}_i^T \mathbf{w}_j|}{\|\mathbf{W}_i\| \|\mathbf{w}_j\|}$

[3] This happens for instance as a result of the Batch Normalization (Ioffe and Szegedy, 2015) technique, which leads to significantly faster convergence. It is thus a good practice to have a mean centered pre-activation distribution.

[4] depending on the distribution $f_c(.)$

any arbitrary normalized distribution. The only restriction assumption 1 has is that to be bounded. However, this does not change the representative power of this distribution significantly because: a) the distributions used for modeling neurons have very small tail mass; b) in practice, we are generally interested in signals with upper bounded values.

The generation process considered in this section (*i.e.* eq. 3 and assumptions 1) is justified because:
**1.** This data generation model finds applications in a number of areas (Yang et al., 2009; Kavukcuoglu et al., 2010; Wright et al., 2009). Notice that while $\mathbf{x}$ is the measurement vector (observed data), which can in general be noisy, $\mathbf{h}$ denotes the actual signal (internal representation) because it reflects the combination of dictionary ($\mathbf{W}^T$) atoms involved in generating the observed samples and hence serves as the true identity of the data.
**2.** Sparse distributed representation (Hinton, 1984) is both observed and desired in hidden representations. It has been empirically shown that representations that are *truly sparse* (*i.e.* large number of true zeros) and *distributed* usually yield better linear separability and performance (Glorot et al., 2011; Wright et al., 2009; Yang et al., 2009).

**Decoding bias ($\mathbf{b}_d$):** Consider the data generation process (exclude noise for now) $\mathbf{x} = \mathbf{W}^T\mathbf{h} + \mathbf{b}_d$. Here $\mathbf{b}_d$ is a bias vector which can take any arbitrary value but similar to $\mathbf{W}$, it is fixed for any particular data generation process. However, the following remark shows that if an AE can recover the sparse code ($\mathbf{h}$) from a data sample generated as $\mathbf{x} = \mathbf{W}^T\mathbf{h}$, then it is also capable of recovering the sparse code from the data generated as $\mathbf{x} = \mathbf{W}^T\mathbf{h} + \mathbf{b}_d$ and vice versa.

**Remark 1.** *Let* $\mathbf{x}_1 = \mathbf{W}^T\mathbf{h}$ *where* $\mathbf{x}_1 \in \mathbb{R}^n$, $\mathbf{W} \in \mathbb{R}^{m \times n}$ *and* $\mathbf{h} \in \mathbb{R}^m$. *Let* $\mathbf{x}_2 = \mathbf{W}^T\mathbf{h} + \mathbf{b}_d$ *where* $\mathbf{b}_d \in \mathbb{R}^n$ *is a fixed vector. Let* $\hat{\mathbf{h}}_1 = s_e(\mathbf{W}\mathbf{x}_1 + \mathbf{b})$ *and* $\hat{\mathbf{h}}_2 = s_e(\mathbf{W}\mathbf{x}_2 + \mathbf{b} - \mathbf{W}\boldsymbol{b}_d)$. *Then* $\hat{\mathbf{h}}_1 = \mathbf{h}$ *iff* $\hat{\mathbf{h}}_2 = \mathbf{h}$.

Thus without any loss of generality, we will assume our data is generated from $\mathbf{x} = \mathbf{W}^T\mathbf{h} + \mathbf{e}$.

## 4 SIGNAL RECOVERY ANALYSIS

We analyse two separate classes of signals in this category– continuous sparse, and binary sparse signals that follow BINS. For notational convenience, we will drop the subscript of $\mathbf{b}_e$ and simply refer this parameter as $\mathbf{b}$ since it is the only bias vector (we are not considering the other bias $\mathbf{b}_d$ due to remark 1). The *Auto-Encoder signal recovery mechanism* that we analyze throughout this paper is defined as,

**Definition 1.** *Let a data sample* $\mathbf{x} \in \mathbb{R}^n$ *be generated by the process* $\mathbf{x} = \mathbf{W}^T\mathbf{h} + \mathbf{e}$ *where* $\mathbf{W} \in \mathbb{R}^{m \times n}$ *is a fixed matrix,* $\mathbf{e}$ *is noise and* $\mathbf{h} \in \mathbb{R}^m$. *Then we define the **Auto-Encoder signal recovery mechanism** as* $\hat{\mathbf{h}}_{s_e}(\mathbf{x}; \mathbf{W}, \mathbf{b}_e)$ *that recovers the estimate* $\hat{\mathbf{h}} = s_e(\mathbf{W}\mathbf{x} + \mathbf{b}_e)$ *where* $s_e(.)$ *is an activation function.*

### 4.1 BINARY SPARSE SIGNAL ANALYSIS

First we consider the noiseless case of data generation,

**Theorem 1.** *(Noiseless Binary Signal Recovery): Let each element of* $\mathbf{h}$ *follow BINS($\mathbf{p}, \delta_1, \mu_h, \mathbf{l}_{\max}$) and let* $\hat{\mathbf{h}} \in \mathbb{R}^m$ *be an auto-encoder signal recovery mechanism with Sigmoid activation function and bias* $\mathbf{b}$ *for a measurement vector* $\mathbf{x} \in \mathbb{R}^n$ *such that* $\mathbf{x} = \mathbf{W}^T\mathbf{h}$. *If we set* $b_i = -\sum_j a_{ij}p_j \; \forall i \in [m]$, *then* $\forall \, \delta \in (0, 1)$,

$$\Pr\left(\frac{1}{m}\|\hat{\mathbf{h}} - \mathbf{h}\|_1 \le \delta\right) \ge 1 - \sum_{i=1}^m \left((1 - p_i)e^{-2\frac{(\delta' + p_i a_{ii})^2}{\sum_{j=1, j \ne i}^m a_{ij}^2}} + p_i e^{-2\frac{(\delta' + (1 - p_i)a_{ii})^2}{\sum_{j=1, j \ne i}^m a_{ij}^2}}\right)$$

(5)

*where* $a_{ij} = \mathbf{W}_i^T\mathbf{W}_j$, $\delta' = \ln(\frac{\delta}{1-\delta})$ *and* $\mathbf{W}_i$ *is the* $i^{th}$ *row of the matrix* $\mathbf{W}$ *cast as a column vector.*

**Analysis**: We first analyse the properties of the weight matrix $\mathbf{W}$ that results in strong recovery bound. Notice the terms $(\delta' + p_i a_{ii})^2$ and $(\delta' + (1 - p_i)a_{ii})^2$ need to be as large as possible, while simultaneously, the term $\sum_{j=1, j \ne i}^m a_{ij}^2$ needs to be as close to zero as possible. For the sake of analysis, lets set[5] $\delta' = 0$ (achieved when $\delta = 0.5$). Then our problem gets reduced to maximizing the

---

[5]Setting $\delta = 0.5$ is not such a bad choice after all because for binary signals, we can recover the exact true signal with high probability by simply binarize the signal recovered by Sigmoid with some threshold.

ratio $\frac{(a_{ii})^2}{\sum_{j=1,j\neq i}^m a_{ij}^2} = \frac{\|\mathbf{W}_i\|^4}{\sum_{j=1,j\neq i}^m (\mathbf{W}_i^T\mathbf{W}_j)^2} = \frac{\|\mathbf{W}_i\|^2}{\sum_{j=1,j\neq i}^m \|\mathbf{W}_j\|^2 \cos^2_{\theta_{ij}}}$, where $\theta_{ij}$ is the angle between $\mathbf{W}_i$ and $\mathbf{W}_j$. From the property of coherence, if the rows of the weight matrix are highly incoherent, then $\cos\theta_{ij}$ is close to 0. Again, for the ease of analysis, lets replace each $\cos\theta_{ij}$ with a small positive number $\epsilon$. Then $\frac{(a_{ii})^2}{\sum_{j=1,j\neq i}^m a_{ij}^2} \approx \frac{\|\mathbf{W}_i\|^2}{\epsilon^2 \sum_{j=1,j\neq i}^m \|\mathbf{W}_j\|^2} = \frac{1}{\epsilon^2 \sum_{j=1,j\neq i}^m \|\mathbf{W}_j\|^2/\|\mathbf{W}_i\|^2}$. Finally, since we would want this term to be maximized for each hidden unit $h_i$ equally, the obvious choice for each weight length $\|\mathbf{W}_i\|$ ($i \in [m]$) is to set it to 1.

Finally, lets analyse the bias vector. Notice we have instantiated each element of the encoding bias $b_i$ to take value $-\sum_j a_{ij}p_j$. Since $p_j$ is essentially the mean of each binary hidden unit $h_i$, we can say that $b_i = -\sum_j a_{ij}\mathbb{E}_{h_j}[h_j] = -\mathbf{W}_i^T\mathbf{W}^T\mathbb{E}_{\mathbf{h}}[\mathbf{h}] = -\mathbf{W}_i^T\mathbb{E}_{\mathbf{h}}[\mathbf{x}]$.

---

Signal recovery is strong for binary signals when the recovery mechanism is given by

$$\hat{h}_i \triangleq \text{Sigmoid}(\mathbf{W}_i^T(\mathbf{x} - \mathbb{E}_{\mathbf{h}}[\mathbf{x}])) \tag{6}$$

where the rows of $\mathbf{W}$ are highly incoherent and each hidden weight has length ones ($\|\mathbf{W}_i\|_2 = 1$), and each dimension of data $\mathbf{x}$ is approximately uncorrelated (see theorem 3).

---

Now we state the recovery bound for the noisy data generation scenario.

**Proposition 1.** *(Noisy Binary Signal Recovery): Let each element of $\mathbf{h}$ follow BINS($\mathbf{p}, \delta_1, \mu_h, \mathbf{l}_{\max}$) and let $\hat{\mathbf{h}} \in \mathbb{R}^m$ be an auto-encoder signal recovery mechanism with Sigmoid activation function and bias $\mathbf{b}$ for a measurement vector $\mathbf{x} = \mathbf{W}^T\mathbf{h} + \mathbf{e}$ where $e \in \mathbb{R}^n$ is any noise vector independent of $\mathbf{h}$. If we set $b_i = -\sum_j a_{ij}p_j - \mathbf{W}_i^T\mathbb{E}_{\mathbf{e}}[\mathbf{e}] \; \forall i \in [m]$, then $\forall \delta \in (0,1)$,*

$$\Pr\left(\frac{1}{m}\|\hat{\mathbf{h}} - \mathbf{h}\|_1 \leq \delta\right) \geq 1 - \sum_{i=1}^m \left((1-p_i)e^{-2\frac{(\delta'-\mathbf{W}_i^T(\mathbf{e}-\mathbb{E}_{\mathbf{e}}[\mathbf{e}])+p_i a_{ii})^2}{\sum_{j=1,j\neq i}^m a_{ij}^2}}\right. \tag{7}$$

$$\left. +p_i e^{-2\frac{(\delta'-\mathbf{W}_i^T(\mathbf{e}-\mathbb{E}_{\mathbf{e}}[\mathbf{e}])+(1-p_i)a_{ii})^2}{\sum_{j=1,j\neq i}^m a_{ij}^2}}\right) \tag{8}$$

*where $a_{ij} = \mathbf{W}_i^T\mathbf{W}_j$, $\delta' = \ln(\frac{\delta}{1-\delta})$ and $\mathbf{W}_i$ is the $i^{th}$ row of the matrix $\mathbf{W}$ cast as a column vector.*

We have not assumed any distribution on the noise random variable $\mathbf{e}$ and this term has no effect on recovery (compared to the noiseless case) if the noise distribution is orthogonal to the hidden weight vectors. Again, the same properties of $\mathbf{W}$ lead to better recovery as in the noiseless case. In the case of bias, we have set each element of the bias $b_i \triangleq -\sum_j a_{ij}p_j - \mathbf{W}_i^T\mathbb{E}_{\mathbf{e}}[\mathbf{e}] \; \forall i \in [m]$. Notice from the definition of BINS, $\mathbb{E}_{h_j}[h_j] = p_j$. Thus in essence, $b_i = -\sum_j a_{ij}\mathbb{E}_{h_j}[h_j] - \mathbf{W}_i^T\mathbb{E}_{\mathbf{e}}[\mathbf{e}]$. Expanding $a_{ij}$, we get, $b_i \triangleq -\mathbf{W}_i^T\mathbf{W}^T\mathbb{E}_{\mathbf{h}}[\mathbf{h}] - \mathbf{W}_i^T\mathbb{E}_{\mathbf{e}}[\mathbf{e}] = -\mathbf{W}_i^T\mathbb{E}_{\mathbf{h}}[\mathbf{x}]$. Thus the expression of bias is unaffected by error statistics as long as we can compute the data mean.

In this section, we will first consider the case when data ($\mathbf{x}$) is generated by linear process $\mathbf{x} = \mathbf{W}^T\mathbf{h} + \mathbf{e}$, and if $\mathbf{W}$ and encoding bias $\mathbf{b}$ have certain properties, then the signal recovery bound ($\|\mathbf{h} - \hat{\mathbf{h}}\|$) is strong. We will then consider the case when data generated by a non-linear process $\mathbf{x} = s_d(\mathbf{W}^T\mathbf{h} + \mathbf{b}_d + \mathbf{e})$ (for certain class of functions $s_d(.)$) can be recovered as well by the same mechanism. For deep non-linear networks, this means that forward propagating data to hidden layers, such that the network parameters satisfy the required conditions, implies each hidden layer recovers the *true* signal that generated the corresponding data. We have moved all the proofs to appendix for better readability.

## 4.2 CONTINUOUS SPARSE SIGNAL RECOVERY

**Theorem 2.** *(Noiseless Continuous Signal Recovery): Let each element of $\mathbf{h} \in \mathbb{R}^m$ follow BINS($\mathbf{p}, f_c, \mu_{\mathbf{h}}, \mathbf{l}_{\max}$) distribution and let $\hat{\mathbf{h}}_{ReLU}(\mathbf{x}; \mathbf{W}, \mathbf{b})$ be an auto-encoder signal recovery mechanism with Rectified Linear activation function (ReLU) and bias $\mathbf{b}$ for a measurement vector*

$\mathbf{x} \in \mathbb{R}^n$ *such that* $\mathbf{x} = \mathbf{W}^T\mathbf{h}$. ***If we set*** $b_i \triangleq -\sum_j a_{ij}p_j\mu_{h_j} \ \forall i \in [m]$, ***then*** $\forall \ \delta \geq 0$,

$$\Pr\left(\frac{1}{m}\|\hat{\mathbf{h}} - \mathbf{h}\|_1 \leq \delta\right) \geq 1 - \sum_{i=1}^m \left( e^{-2\frac{(\delta+\sum_j(1-p_j)(l_{\max_j}-2p_j\mu_{h_j})\max(0,a_{ij}))^2}{\sum_j a_{ij}^2 l_{\max_j}^2}} \right.$$
$$\left. + e^{-2\frac{(\delta+\sum_j(1-p_j)(l_{\max_j}-2p_j\mu_{h_j})\max(0,-a_{ij}))^2}{\sum_j a_{ij}^2 l_{\max_j}^2}} \right) \tag{9}$$

*where* $\mathbf{a}_i s$ *are vectors such that*

$$a_{ij} = \begin{cases} \mathbf{W}_i^T\mathbf{W}_j & if \quad i \neq j \\ \mathbf{W}_i^T\mathbf{W}_i - 1 & if \quad i = j \end{cases} \tag{10}$$

$\mathbf{W}_i$ *is the* $i^{th}$ *row of the matrix* $\mathbf{W}$ *cast as a column vector.*

**Analysis**: We first analyze the properties of the weight matrix that results in strong recovery bound. We find that for strong recovery, the terms $(\delta + \sum_j(1 - p_j)(l_{\max_j} - 2p_j\mu_{h_j})\max(0, a_{ij}))^2$ and $(\delta + \sum_j(1-p_j)(l_{\max_j} - 2p_j\mu_{h_j})\max(0, -a_{ij}))^2$ should be as large as possible, while simultaneously, the term $\sum_j a_{ij}^2 l_{\max_j}^2$ needs to be as close to zero as possible. First, notice the term $(1 - p_j)(l_{\max_j} - 2p_j\mu_{h_j})$. Since $\mu_{h_j} < l_{\max_j}$ by definition, we have that both terms containing $(1 - p_j)(l_{\max_j} - 2p_j\mu_{h_j})$ are always positive and contributes towards stronger recovery if $p_j$ is less than $50\%$ (sparse), and becomes stronger as the signal becomes sparser (smaller $p_j$).

Now if we assume the rows of the weight matrix $\mathbf{W}$ are highly incoherent and that each row of $\mathbf{W}$ has unit $\ell^2$ length, then it is safe to assume each $a_{ij}$ ($\forall i, j \in [m]$) is close to 0 from the definition of $a_{ij}$ and properties of $\mathbf{W}$ we have assumed. Then for any small positive value of $\delta$, we can approximately say $\frac{(\delta + \sum_j(1-p_j)(l_{\max_j}-2p_j\mu_{h_j})\max(0,a_{ij}))^2}{\sum_j a_{ij}^2 l_{\max_j}^2} \approx \frac{\delta^2}{\sum_j a_{ij}^2 l_{\max_j}^2}$ where each $a_{ij}$ is very close to zero. The same argument holds similarly for the other term. Thus we find that a strong signal recovery bound would be obtained if the weight matrix is highly incoherent and all hidden vectors are of unit length.

In the case of bias, we have set each element of the bias $b_i \triangleq -\sum_j a_{ij}p_j\mu_{h^j} \ \forall i \in [m]$. Notice from the definition of BINS, $\mathbb{E}_{h_j}[h_j] = p_j\mu_{h_j}$. Thus in essence, $b_i = -\sum_j a_{ij}\mathbb{E}_{h_j}[h_j]$. Expanding $a_{ij}$, we get $b_i = -\mathbf{W}_i^T\mathbf{W}^T\mathbb{E}_\mathbf{h}[\mathbf{h}] + \mathbb{E}_{h_i}[h_i] = -\mathbf{W}_i^T\mathbb{E}_\mathbf{h}[\mathbf{x}] + \mathbb{E}_{h_i}[h_i]$.

---

The recovery bound is strong for continuous signals when the recovery mechanism is set to

$$\hat{h}_i \triangleq \text{ReLU}(\mathbf{W}_i^T(\mathbf{x} - \mathbb{E}_\mathbf{x}[\mathbf{x}]) + \mathbb{E}_{h_i}[h_i]) \tag{11}$$

and the rows of $\mathbf{W}$ are highly incoherent and each hidden weight has length ones ($\|\mathbf{W}_i\|_2 = 1$).

---

Now we state the recovery bound for the noisy data generation scenario.

**Proposition 2.** *(Noisy Continuous Signal Recovery): Let each element of* $\mathbf{h} \in \mathbb{R}^m$ *follow* BINS$(\mathbf{p}, f_c, \mu_\mathbf{h}, \mathbf{l}_{\max})$ *distribution and let* $\hat{\mathbf{h}}_{ReLU}(\mathbf{x}; \mathbf{W}, \mathbf{b})$ *be an auto-encoder signal recovery mechanism with Rectified Linear activation function (ReLU) and bias* $\mathbf{b}$ *for a measurement vector* $\mathbf{x} \in \mathbb{R}^n$ *such that* $\mathbf{x} = \mathbf{W}^T\mathbf{h} + \mathbf{e}$ *where* $\mathbf{e}$ *is any noise random vector independent of* $\mathbf{h}$. ***If we set*** $b_i \triangleq -\sum_j a_{ij}p_j\mu_{h_j} - \mathbf{W}_i^T\mathbb{E}_\mathbf{e}[\mathbf{e}] \ \forall i \in [m]$, ***then*** $\forall \ \delta \geq 0$,

$$\Pr\left(\frac{1}{m}\|\hat{\mathbf{h}} - \mathbf{h}\|_1 \leq \delta\right) \geq 1 - \sum_{i=1}^m \left( e^{-2\frac{(\delta-\mathbf{W}_i^T(\mathbf{e}-\mathbb{E}_\mathbf{e}[\mathbf{e}])+\sum_j(1-p_j)(l_{\max_j}-2p_j\mu_{h_j})\max(0,a_{ij}))^2}{\sum_j a_{ij}^2 l_{\max_j}^2}} \right.$$
$$\left. + e^{-2\frac{(\delta-\mathbf{W}_i^T(\mathbf{e}-\mathbb{E}_\mathbf{e}[\mathbf{e}])+\sum_j(1-p_j)(l_{\max_j}-2p_j\mu_{h_j})\max(0,-a_{ij}))^2}{\sum_j a_{ij}^2 l_{\max_j}^2}} \right) \tag{12}$$

*where* $\mathbf{a}_i s$ *are vectors such that*

$$a_{ij} = \begin{cases} \mathbf{W}_i^T\mathbf{W}_j & if \quad i \neq j \\ \mathbf{W}_i^T\mathbf{W}_i - 1 & if \quad i = j \end{cases} \tag{13}$$

$\mathbf{W}_i$ is the $i^{th}$ row of the matrix $\mathbf{W}$ cast as a column vector.

Notice that we have not assumed any distribution on variable $\mathbf{e}$, which denotes the noise. Also, this term has no effect on recovery (compared to the noiseless case) if the noise distribution is orthogonal to the hidden weight vectors. On the other hand, the same properties of $\mathbf{W}$ lead to better recovery as in the noiseless case. However, in the case of bias, we have set each element of the bias $b_i \triangleq -\sum_j a_{ij} p_j \mu_{h^j} - \mathbf{W}_i^T \mathbb{E}_\mathbf{e}[\mathbf{e}] \ \forall i \in [m]$. From the definition of BINS, $\mathbb{E}_{h_j}[h_j] = p_j \mu_{h_j}$. Thus $b_i = -\sum_j a_{ij} \mathbb{E}_{h_j}[h_j] - \mathbf{W}_i^T \mathbb{E}_\mathbf{e}[\mathbf{e}]$. Expanding $a_{ij}$, we get, $b_i \triangleq -\mathbf{W}_i^T \mathbf{W}^T \mathbb{E}_\mathbf{h}[\mathbf{h}] + \mathbb{E}_{h_i}[h_i] - \mathbf{W}_i^T \mathbb{E}_\mathbf{e}[\mathbf{e}] = -\mathbf{W}_i^T \mathbb{E}_\mathbf{h}[\mathbf{x}] + \mathbb{E}_{h_i}[h_i]$. Thus the expression of bias is unaffected by error statistics as long as we can compute the data mean (*i.e.* the recovery is the same as shown in eq. 11).

## 4.3 PROPERTIES OF GENERATED DATA

Since the data we observe results from the hidden signal given by $\mathbf{x} = \mathbf{W}^T \mathbf{h}$, it would be interesting to analyze the distribution of the generated data. This would provide us more insight into what kind of pre-processing would ensure stronger signal recovery.

**Theorem 3.** *(Uncorrelated Distribution Bound): If data is generated as $\mathbf{x} = \mathbf{W}^T \mathbf{h}$ where $\mathbf{h} \in \mathbb{R}^m$ has covariance matrix $\mathrm{diag}(\zeta)$, $(\zeta \in \mathbb{R}^{+^m})$ and $\mathbf{W} \in \mathbb{R}^{m \times n}$ $(m > n)$ is such that each row of $\mathbf{W}$ has unit length and the rows of $\mathbf{W}$ are maximally incoherent, then the covariance matrix of the generated data is approximately spherical (uncorrelated) satisfying,*

$$\min_\alpha \|\mathbf{\Sigma} - \alpha \mathbf{I}\|_F \leq \sqrt{\frac{1}{n}\left(m\|\zeta\|_2^2 - \|\zeta\|_1^2\right)} \tag{14}$$

*where $\mathbf{\Sigma} = \mathbb{E}_\mathbf{x}[(\mathbf{x} - \mathbb{E}_\mathbf{x}[\mathbf{x}])(\mathbf{x} - \mathbb{E}_\mathbf{x}[\mathbf{x}])^T]$ is the covariance matrix of the generated data.*

**Analysis**: Notice that for any vector $\mathbf{v} \in \mathbb{R}^m$, $m\|\mathbf{v}\|_2^2 \geq \|\mathbf{v}\|_1^2$, and the equality holds when each element of the vector $\mathbf{v}$ is identical.

---

Data $\mathbf{x}$ generated using a maximally incoherent dictionary $\mathbf{W}$ (with unit $\ell^2$ row length) as $\mathbf{x} = \mathbf{W}^T \mathbf{h}$ guarantees $\mathbf{x}$ is highly uncorrelated if $\mathbf{h}$ is uncorrelated with near identity covariance. This would ensure the hidden units at the following layer are also uncorrelated during training. Further the covariance matrix of $\mathbf{x}$ is identity, if all hidden units have equal variance.

---

This analysis acts as a justification for data whitening where data is processed to have zero mean and identity covariance matrix. Notice that although the generated data does not have zero mean, the recovery process (eq. 11) subtracts data mean and hence it does not affect recovery.

## 4.4 CONNECTIONS WITH EXISTING WORK

**Auto-Encoders (AE)**: Our analysis reveals the conditions on parameters of an AE that lead to strong recovery of $\mathbf{h}$ (for both continuous and binary case), which ultimately implies low data reconstruction error.

However, the above arguments hold for AEs from a recovery point of view. Training an AE on data may lead to learning of the identity function. Thus usually AEs are trained along with a bottle-neck to make the learned representation useful. One such bottle-neck is the de-noising criteria given by,

$$\mathcal{J}_{DAE} = \min_{\mathbf{W}, \mathbf{b}} \|\mathbf{x} - \mathbf{W}^T s_e(\mathbf{W}\tilde{\mathbf{x}} + \mathbf{b})\|^2 \tag{15}$$

where $s_e(.)$ is the activation function and $\tilde{\mathbf{x}}$ is a corrupted version of $\mathbf{x}$. It has been shown that the Taylor's expansion of DAE (Theorem 3 of Arpit et al., 2016) has the term $\sum_{\substack{j,k=1 \\ j \neq k}}^m \left(\frac{\partial h_j}{\partial a_j} \frac{\partial h_k}{\partial a_k}(\mathbf{W}_j^T \mathbf{W}_k)^2\right)$.

If we constrain the lengths of the weight vectors to have fixed length, then this regularization term minimizes a weighted sum of cosine of the angle between every pair of weight vectors. As a result, the weight vectors become increasingly incoherent. Hence we achieve both our goals by adding one additional constraint to DAE– constraining weight vectors to have unit length. Even if we do not apply an explicit constraint, we can expect the weight lengths to be upper bounded from the basic AE objective itself, which would explain the learning of incoherent weights due to the DAE regularization.On a side note, our analysis also justifies the use of tied weights in auto-encoders.

**Sparse Coding (SC)**: SC involves minimizing $\|\mathbf{x} - \mathbf{W}^T\mathbf{h}\|^2$ using the sparsest possible $\mathbf{h}$. The analysis after theorem 2 shows signal recovery using the AE mechanism becomes stronger for sparser signals (as also confirmed experimentally in section 5). In other words, for any given data sample and weight matrix, as long as the conditions on the weight matrix and bias are met, the AE recovery mechanism recovers the sparsest possible signal; which justifies using auto-encoders for recovering sparse codes (see Henaff et al., 2011; Makhzani and Frey, 2013; Ng, 2011, for work along this line).

**Independent Component Analysis**(ICA): ICA (Hyvärinen and Oja, 2000; Arora et al., 2012) assumes we observe data generated by the process $\mathbf{x} = \mathbf{W}^T\mathbf{h}$ where all elements of the $\mathbf{h}$ are independent and $\mathbf{W}$ is a mixing matrix. The task of ICA is to recover both $\mathbf{W}$ and $\mathbf{h}$ given data. This data generating process is precisely what we assumed in section 3. Based on this assumption, our results show that **1)** the properties of $\mathbf{W}$ that can recover such independent signals $\mathbf{h}$; and **2)** auto-encoders can be used for recovering such signals and weight matrix $\mathbf{W}$.

**k-Sparse AEs** : Makhzani and Frey (2013) propose to zero out all the values of hidden units smaller than the top-k values for each sample during training. This is done to achieve sparsity in the learned hidden representation. This strategy is justified from the perspective of our analysis as well. This is because the PAC bound (theorem 2) derived for signal recovery using the AE signal recovery mechanism shows we recover a noisy version of the true sparse signal. Since the noise in each recovered signal unit is roughly proportional to the original value, de-noising such recovered signals can be achieved by thresholding the hidden unit values (exploiting the fact that the signal is sparse). This can be done either by using a fixed threshold or picking the top k values.

**Data Whitening**: Theorem 3 shows that data generated from BINS and incoherent weight matrices are roughly uncorrelated. Thus recovering back such signals using auto-encoders would be easier if we pre-process the data to have uncorrelated dimensions.

## 5 EMPIRICAL VERIFICATION

We empirically verify the fundamental predictions made in section 4 which both serve to justify the assumptions we have made, as well as confirm our results. We verify the following: a) the optimality of the rows of a weight matrix $\mathbf{W}$ to have unit length and being highly incoherent for AE signal recovery; b) effect of sparsity on AE signal recovery; and c) in practice, AE can recover not only the true sparse signal $\mathbf{h}$, but also the dictionary $\mathbf{W}$ that used to generate the data.

### 5.1 OPTIMAL PROPERTIES OF WEIGHTS AND BIAS

Our analysis on signal recovery in section 4 (eq. 11) shows signal recovery bound is strong when a) the data generating weight matrix $\mathbf{W}$ has rows of unit $\ell^2$ length; b) the rows of $\mathbf{W}$ are highly incoherent; c) each bias vector element is set to the negative expectation of the pre-activation; d) signal $\mathbf{h}$ has each dimension independent. In order to verify this, we generate $N = 5,000$ signals $\mathbf{h} \in \mathbb{R}^{m=200}$ from BINS($p$=0.02,$f_c$=uniform, $\mu_h$=0.5,$l_{\max}$=1) with $f_c(.)$ set to uniform distribution for simplicity. We then generate the corresponding $5,000$ data sample $\mathbf{x} = c\mathbf{W}^T\mathbf{h} \in \mathbb{R}^{180}$ using an incoherent weight matrix $\mathbf{W} \in \mathbb{R}^{200 \times 180}$ (each element sampled from zero mean Gaussian, the columns are then orthogonalized, and $\ell^2$ length of each row rescaled to 1; notice the rows cannot be orthogonal). We then recover each signal using,

$$\hat{h}_i \triangleq \text{ReLU}(c\mathbf{W}_i^T(\mathbf{x} - \mathbb{E}_\mathbf{h}[\mathbf{x}]) + \mathbb{E}_{h_i}[h_i] + \Delta b) \tag{16}$$

where $c$ and $\Delta b$ are scalars that we vary between $[0.1, 2]$ and $[-1, +1]$ respectively. We also generate $N = 5,000$ signals $\mathbf{h} \in \{0,1\}^{m=200}$ from BINS($0.02,\delta_1$, $0.02,1$) with $f_c(.)$ set to Dirac delta function at 1. We then generate the corresponding $5,000$ data sample $\mathbf{x} = c\mathbf{W}^T\mathbf{h} \in \mathbb{R}^{180}$ following the same procedure as for the continuous signal case. The signal is recovered using

$$\hat{h}_i \triangleq \sigma(c\mathbf{W}_i^T(\mathbf{x} - \mathbb{E}_\mathbf{h}[\mathbf{x}]) + \Delta b) \tag{17}$$

where $\sigma$ is the sigmoid function. For the recovered signals, we calculate the *Average Percentage Recovery Error* (APRE) as,

$$\text{APRE} = \frac{100}{Nm} \sum_{i=1,j=1}^{N,m} w_{h_j^i} \mathbf{1}(|\hat{h}_j^i - h_j^i| > \epsilon) \tag{18}$$

where we set $\epsilon$ to 0.1 for continuous signals and 0 for binary case, $\mathbf{1}(.)$ is the indicator operator, $\hat{h}_j^i$ denotes the $j^{th}$ dimension of the recovered signal corresponding to the $i^{th}$ true signal and,

$$
w_{h_j^i} = \begin{cases} \frac{0.5}{p} & if & h_j^i > 0 \\ \frac{0.5}{1-p} & if & h_j^i = 0 \end{cases} \tag{19}
$$

The error is weighted with $w_{h_j^i}$ so that the recovery error for both zero and non-zero $h_j^i$s are penalized equally. This is specially needed in this case, because $h_j^i$ is sparse and a low error can also be achieved by trivially setting all the recovered $\hat{h}_j^i$s to zero. Along with the incoherent weight matrix, we also generate data separately using a highly coherent weight matrix that we get by sampling each element randomly from a uniform distribution on $[0, 1]$ and scaling each row to unit length. According to our analysis, we should get least error for $c = 1$ and $\Delta b = 0$ for the incoherent matrix while the coherent matrix should yield both higher recovery error and a different choice of $c$ and $b$ (which is unpredictable). The error heat maps for both continuous and binary recovery[6] are shown in fig. 1. For the incoherent weight matrix, we see that the empirical optimal is precisely $c = 1$ and $\Delta b = 0$ (which is exactly as predicted) with 0.21 and 0.0 APRE for continuous and binary recovery, respectively. It is interesting to note that the binary recovery is quite robust with the choice of $c$ and $\Delta b$, which is because 1) the recovery is denoised through thresholding, and 2) the binary signal inherently contains less information and thus is easier to recover. For the coherent weight matrix, we get 45.75 and 32.63 APRE instead (see fig. 5).

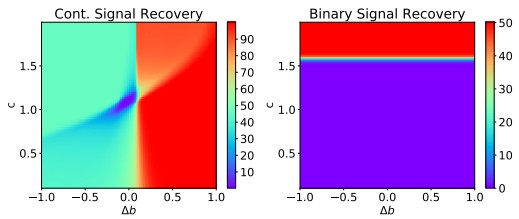

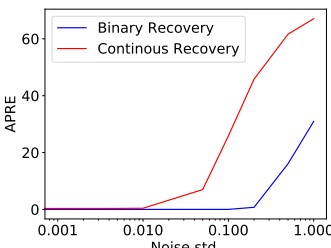

Figure 1: Error heatmap showing optimal values of $c$ and $\Delta b$ for recovering continous (left) and binary (right) signal using inchoherent weights.

Figure 2: Average percentage recovery error of noisy signal recovery.

We also experiment on the noisy recovery case, where we generate the data using incoherent weight matrix with $c = 1$ and $\Delta b = 0$. For each data dimension we add independent Gaussian noise with mean 100 with standard deviation varying from 0.001 to 1. Both signal recovery schemes are quite robust against noise (see fig. 2). In particular, the binary signal recovery is very robust, which conforms with our previous observation.

## 5.2 EFFECT OF SPARSITY ON SIGNAL RECOVERY

We analyze the effect of sparsity of signals on their recovery using the mechanism shown in section 4. In order to do so, we generate incoherent matrices using two different methods– Gaussian[7] and orthogonal (Saxe et al., 2013). In addition, all the generated weight matrices are normalized to have unit $\ell^2$ row length. We then sample signals and generate data using the same configurations as mentioned in section 5.1; only this time, we fix $c = 1$ and $\Delta b = 0$, vary hidden unit activation probability $p$ in $[0.02, 1]$, and duplicate the generated data while adding noise to the copy, which we sample from a Gaussian distribution with mean 100 and standard deviation 0.05. According to our analysis, noise mean should have no effect on recovery so the mean value of 100 shouldn't have any effect; only standard deviation affects recovery. We find for all weight matrices, recovery error reduces with increasing sparsity (decreasing $p$, see fig. 3). Additionally, we find that both recovery are robust against noise. We also find the recovery error trend is almost always lower for orthogonal weight matrices, especially when the signal is sparse. [8] Recall theorem 2 suggests stronger recovery for more incoherent matrices. So we look into the row coherence of $\mathbf{W} \in \mathbb{R}^{m \times n}$ sampled from Gaussian and Orthogonal methods with $m = 200$ and varying $n \in [100, 300]$. We found that

---

[6]We use 0.55 as the threshold to binarize the recovered signal using sigmoid function.

[7]Gaussian and Xavier (Glorot and Bengio, 2010) initialization becomes identical after weight length normalization

[8]notice the rows of $\mathbf{W}$ are not orthogonal for overcomplete filters, rather the columns are orthogonalized, unless $\mathbf{W}$ is undercomplete

orthogonal initialized matrices have significantly lower coherence even though the orthogonalization is done column-wise (see fig. 6.). This explains significantly lower recovery error for orthogonal matrices in figure 3.

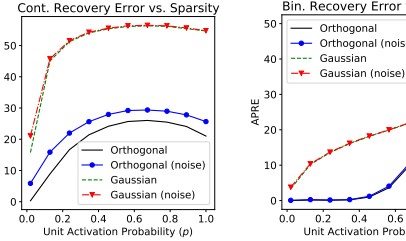
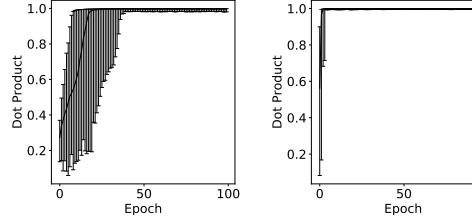

Figure 3: Effect of signal sparseness on continuous (left) and binary (right) signal recovery. Noise in parenthesis indicate the generated data was corrupted with Gaussian noise. Sparser signals are recovered better.

Figure 4: Cosine similarity between greedy paired rows of $\mathbf{W}$ and $\hat{\mathbf{W}}$ for continuous (left) and binary (right) recovery. The upper, mid and lower bar denotes the 95th, 50th and 5th percentile.

## 5.3 RECOVERY OF DATA DICTIONARY

We showed the conditions on $\mathbf{W}$ and $\mathbf{b}$ for good recovery of sparse signal $\mathbf{h}$. In practice, however, one does not have access to $\mathbf{W}$, in general. Therefore, in this section, we empirically demonstrate that AE can indeed recover both $\mathbf{W}$ and $\mathbf{h}$ through optimizing the AE objective. We generate $50,000$ signals $\mathbf{h} \in \mathbb{R}^{m=200}$ with the same BINS distribution as in section 5.1. The data are then generated as $\mathbf{x} = \mathbf{W}^T \mathbf{h}$ using an incoherent weight matrix $\mathbf{W} \in \mathbb{R}^{200 \times 180}$ (same as in section 5.1). We then recover the data dictionary $\hat{\mathbf{W}}$ by:

$$\hat{\mathbf{W}} = \arg \min_{\mathbf{W}} \mathbb{E}_{\mathbf{x}} \left[ \| \mathbf{x} - \mathbf{W}^T s_e(\mathbf{W}(\mathbf{x} - \mathbb{E}_{\mathbf{x}}[\mathbf{x}])) \|^2 \right], \qquad \text{where} \| \mathbf{W_i} \|_2^2 = 1 \forall i \qquad (20)$$

Notice that although given sparse signal $\mathbf{h}$ the data dictionary $\mathbf{W}$ is unique (Hillar and Sommer, 2015), there are $m!$ number of equivalent solutions for $\hat{\mathbf{W}}$, since we can permute dimension of $\mathbf{h}$ in AE. To check if the original data dictionary is recovered, we therefore pair up the rows of $\mathbf{W}$ and $\hat{\mathbf{W}}$ by greedily selecting the pairs that result in the highest dot product value. We then measure the goodness of the recovery by looking at the values of all the paired dot products. In addition, since we know the pairing, we can calculate APRE to evaluate the quality of recovered hidden signal. As can be observed from fig. 4, by optimizing the AE objective we can recover the original data dictionary $\mathbf{W}$ (almost all of the cosine distances are 1). The final achieved $1.61$ and $0.15$ APRE for continuous and binary signal recovery, which is a bit less than what we achieved in section 5.1. However, one should note that for this set of experiments we only observed data $\mathbf{x}$ and no other information regarding $\mathbf{W}$ is exposed. Not surprisingly, we again observed that the binary signal recovery is more robust as compared to the continuous counterpart, which may attribute to its lower information content. We also did experiments on noisy data and achieved similar performance as in section 5.1 when the noise is less significant (see supplementary materials for more details). These results strongly suggests that AEs are capable of recovering the true hidden signal in practice.

## 6 CONCLUSION

In this paper we looked at the sparse signal recovery problem from the Auto-Encoder perspective and provide novel insights into conditions under which AEs can recover such signals. In particular, 1) from the signal recovery stand point, if we assume that the observed data is generated from some sparse hidden signals according to the assumed data generating process, then, the *true* hidden representation can be approximately recovered if a) the weight matrices are highly incoherent with unit $\ell^2$ row length, and b) the bias vectors are as described in equation 11 (theorem 2)[9]. The recovery also becomes more and more accurate with increasing sparsity in hidden signals. 2) From the data generation perspective, we found that data generated from such signals (assumption 1) have the property of being roughly uncorrelated (theorem 3), and thus pre-processing the data to have

---

[9]For binary recovery, the bias equation is described in 6

uncorrelated dimensions may encourage stronger signal recovery. 3) Given only measurement data, we empirically show that the AE reconstruction objective recovers the data generating dictionary, and hence the true signal $\mathbf{h}$. 4) These conditions and observations allow us to view various existing techniques, such as data whitening, independent component analysis, *etc.*, in a more coherent picture when considering signal recovery.

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

# Appendix: On Optimality Conditions for Auto-Encoder Signal Recovery

## 1 PROOFS

**Remark 1.** *Let* $\mathbf{x}_1 = \mathbf{W}^T \mathbf{h}$ *where* $\mathbf{x}_1 \in \mathbb{R}^n$, $\mathbf{W} \in \mathbb{R}^{m \times n}$ *and* $\mathbf{h} \in \mathbb{R}^m$. *Let* $\mathbf{x}_2 = \mathbf{W}^T \mathbf{h} + \boldsymbol{b}_d$ *where* $\mathbf{b}_d \in \mathbb{R}^n$ *is a fixed vector. Let* $\hat{\mathbf{h}}_1 = s_e(\mathbf{W}\mathbf{x}_1 + \mathbf{b})$ *and* $\hat{\mathbf{h}}_2 = s_e(\mathbf{W}\mathbf{x}_2 + \mathbf{b} - \mathbf{W}\boldsymbol{b}_d)$. *Then* $\hat{\mathbf{h}}_1 = \boldsymbol{h}$ *iff* $\hat{\mathbf{h}}_2 = \boldsymbol{h}$.

*Proof: Let* $\hat{\mathbf{h}}_1 = \mathbf{h}$. *Thus* $\mathbf{h} = s_e(\mathbf{W}\mathbf{x}_1 + \mathbf{b})$. *On the other hand,* $\hat{\mathbf{h}}_2 = s_e(\mathbf{W}\mathbf{x}_2 + \mathbf{b} - \mathbf{W}\boldsymbol{b}_d) = s_e(\mathbf{W}\mathbf{W}^T\mathbf{h} + \mathbf{W}\mathbf{b}_d + \mathbf{b} - \mathbf{W}\boldsymbol{b}_d) = s_e(\mathbf{W}\mathbf{W}^T\mathbf{h} + \mathbf{b}) = s_e(\mathbf{W}\mathbf{x}_1 + \mathbf{b}) = \mathbf{h}$. *The other direction can be proved similarly.*

**Theorem 1.** *Let each element of* $\mathbf{h}$ *follow BINS($\mathbf{p}, \delta_1, \mu_h, \mathbf{l}_{\max}$) and let* $\hat{\mathbf{h}} \in \mathbb{R}^m$ *be an auto-encoder signal recovery mechanism with Sigmoid activation function and bias* $\mathbf{b}$ *for a measurement vector* $\mathbf{x} \in \mathbb{R}^n$ *such that* $\mathbf{x} = \mathbf{W}^T\mathbf{h}$. *If we set* $b_i = -\sum_j a_{ij}p_j \ \forall i \in [m]$, **then** $\forall \ \delta \in (0,1)$,

$$\Pr\left(\frac{1}{m}\|\hat{\mathbf{h}} - \mathbf{h}\|_1 \leq \delta\right) \geq 1 - \sum_{i=1}^m \left((1-p_i)e^{-2\frac{(\delta'+p_i a_{ii})^2}{\sum_{j=1,j\neq i}^m a_{ij}^2}} + p_i e^{-2\frac{(\delta'+(1-p_i)a_{ii})^2}{\sum_{j=1,j\neq i}^m a_{ij}^2}}\right) \quad (21)$$

*where* $a_{ij} = \mathbf{W}_i^T\mathbf{W}_j$, $\delta' = \ln(\frac{\delta}{1-\delta})$ *and* $\mathbf{W}_i$ *is the* $i^{th}$ *row of the matrix* $\mathbf{W}$ *cast as a column vector.*

*Proof.* Notice that,

$$\Pr(|\hat{h}_i - h_i| \geq \delta) = \Pr(|\hat{h}_i - h_i| \geq \delta \,\big|\, h_i = 0)\Pr(h_i = 0) + \Pr(|\hat{h}_i - h_i| \geq \delta \,\big|\, h_i = 1)\Pr(h_i = 1) \quad (22)$$

and from definition 1,

$$\hat{h}_i = \sigma\left(\sum_j a_{ij}h_j + b_i\right) \quad (23)$$

Thus,

$$\Pr(|\hat{h}_i - h_i| \geq \delta) = (1-p_i)\Pr\left(\sigma\left(\sum_j a_{ij}h_j + b_i\right) \geq \delta \,\Big|\, h_i = 0\right)$$

$$+ p_i \Pr\left(\sigma\left(-\sum_j a_{ij}h_j - b_i\right) \geq \delta \,\Big|\, h_i = 1\right) \quad (24)$$

Notice that $\Pr(\sigma(\sum_j a_{ij}h_j + b_i) \geq \delta \,\big|\, h_i = 0) = \Pr(\sum_j a_{ij}h_j + b_i \geq \ln(\frac{\delta}{1-\delta}) \,\big|\, h_i = 0)$. **Let** $z_i = \sum_j a_{ij}h_j + b_i$ and $\delta' = \ln(\frac{\delta}{1-\delta})$. Then, **setting** $b_i = -\mathbb{E}_{\mathbf{h}}[\sum_j a_{ij}h_j] = -\sum_j a_{ij}p_j$, using Chernoff's inequality, **for any** $t > 0$,

$$\Pr(z_i \geq \delta' \,\big|\, h_i = 0) \leq \frac{\mathbb{E}_{\mathbf{h}}[e^{tz_i}]}{e^{t\delta'}} = \frac{\mathbb{E}_{\mathbf{h}}\left[e^{t\sum_{j\neq i} a_{ij}(h_j - p_j) - tp_i a_{ii}}\right]}{e^{t\delta'}}$$

$$= \frac{\mathbb{E}_{\mathbf{h}}\left[\prod_{j\neq i} e^{ta_{ij}(h_j - p_j)}\right]}{e^{t(\delta' + p_i a_{ii})}} = \frac{\prod_{j\neq i}\mathbb{E}_{h_j}\left[e^{ta_{ij}(h_j - p_j)}\right]}{e^{t(\delta' + p_i a_{ii})}} \quad (25)$$

Let $T_j = \mathbb{E}_{h_j}\left[e^{ta_{ij}(h_j - p_j)}\right]$. Then,

$$T_j = (1-p_j)e^{-tp_j a_{ij}} + p_j e^{t(1-p_j)a_{ij}} = e^{-tp_j a_{ij}}(1 - p_j + p_j e^{ta_{ij}}) \quad (26)$$

Let $e^{g(t)} \triangleq T_j$, thus,

$$g(t) = -tp_j a_{ij} + \ln(1 - p_j + p_j e^{ta_{ij}}) \implies g(0) = 0 \tag{27}$$

$$g'(t) = -p_j a_{ij} + \frac{p_j a_{ij} e^{ta_{ij}}}{1 - p_j + p_j e^{ta_{ij}}} \implies g'(0) = 0 \tag{28}$$

$$g''(t) = \frac{p_j(1 - p_j)a_{ij}^2 e^{ta_{ij}}}{(1 - p_j + p_j e^{ta_{ij}})^2} \tag{29}$$

$$g'''(t) = \frac{p_j(1 - p_j)a_{ij}^3 e^{ta_{ij}}(1 - p_j + p_j e^{ta_{ij}})(1 - p_j - p_j e^{ta_{ij}})}{(1 - p_j + p_j e^{ta_{ij}})^4} \tag{30}$$

$$\tag{31}$$

Setting $g'''(t) = 0$, we get $t^* = \frac{1}{a_{ij}}\ln(\frac{1-p_j}{p_j})$. Thus, $g''(t) \leq g(t^*) = \frac{a_{ij}^2}{4}$. By Taylor's theorem, $\exists c \in [0, t] \forall t > 0$ s.t.,

$$g(t) = g(0) + tg'(0) + \frac{t^2}{2}g''(c) \leq \frac{t^2 a_{ij}^2}{8} \tag{32}$$

Thus we can upper bound $T_j$ as,

$$T_j \leq e^{\frac{t^2 a_{ij}^2}{8}} \tag{33}$$

Hence we can write $\Pr(z_i \geq \delta')$ as

$$\Pr(z_i \geq \delta') \leq \frac{\prod_{j \neq i} T_j}{e^{t(\delta' + a_{ii} p_i)}} = \frac{\prod_{j \neq i} e^{\frac{t^2 a_{ij}^2}{8}}}{e^{t(\delta' + a_{ii} p_i)}} = e^{\frac{t^2 \sum_{j \neq i} a_{ij}^2}{8} - t(a_{ii} p_i + \delta')} \tag{34}$$

On the other hand, notice $\Pr(\sigma(-\sum_j a_{ij}h_j - b_i) \geq \delta \mid h_i = 1) = \Pr(-\sum_j a_{ij}h_j - b_i \geq \ln(\frac{\delta}{1-\delta}) \mid h_i = 1) = \Pr(-z_i \geq \delta' \mid h_i = 1)$.

$$\Pr(-z_i \geq \delta' \mid h_i = 1) \leq \frac{\mathbb{E}_{\mathbf{h}}[e^{-tz_i}]}{e^{t\delta'}} = \frac{\mathbb{E}_{\mathbf{h}}\left[e^{-t\sum_{j \neq i} a_{ij}(h_j - p_j) - t(1-p_i)a_{ii}}\right]}{e^{t\delta'}} \tag{35}$$

$$= \frac{\mathbb{E}_{\mathbf{h}}\left[\prod_{j \neq i} e^{-ta_{ij}(h_j - p_j)}\right]}{e^{t(\delta' + (1-p_i)a_{ii})}} \tag{36}$$

$$= \frac{\prod_{j \neq i} \mathbb{E}_{h_j}\left[e^{-ta_{ij}(h_j - p_j)}\right]}{e^{t(\delta' + (1-p_i)a_{ii})}} \tag{37}$$

Let $T_j = \mathbb{E}_{h_j}\left[e^{-ta_{ij}(h_j - p_j)}\right]$. Then we can similarly bound $\Pr(-z_i \geq \delta')$ by effectively flipping the sign of $a_{ij}$'s in the previous derivation,

$$\Pr(-z_i \geq \delta') \leq \frac{\prod_{j \neq i} T_j}{e^{t(\delta' + a_{ii}(1-p_i))}} = \frac{\prod_{j \neq i} e^{\frac{t^2 a_{ij}^2}{8}}}{e^{t(\delta' + a_{ii}(1-p_i))}} = e^{\frac{t^2 \sum_{j \neq i} a_{ij}^2}{8} - t(a_{ii}(1-p_i) + \delta')} \tag{38}$$

Minimizing both 34 and 38 with respect to $t$ and applying union bound, we get,

$$\Pr(|\hat{h}_i - h_i| \geq \delta) \leq (1 - p_i)e^{\frac{-2(a_{ii} p_i + \delta')^2}{\sum_{j \neq i} a_{ij}^2}} + p_i e^{\frac{-2(a_{ii}(1-p_i) + \delta')^2}{\sum_{j \neq i} a_{ij}^2}} \tag{39}$$

Since the above bound holds for all $i \in [m]$, applying union bound on all the units yields the desired result. $\qquad\square$

**Proposition 1.** *Let each element of $\mathbf{h}$ follow BINS($\mathbf{p}, \delta_1, \mu_h, \mathbf{l}_{\max}$) and let $\hat{\mathbf{h}} \in \mathbb{R}^m$ be an autoencoder signal recovery mechanism with Sigmoid activation function and bias $\mathbf{b}$ for a measurement*

*vector* $\mathbf{x} = \mathbf{W}^T\mathbf{h} + \mathbf{e}$ *where* $e \in \mathbb{R}^n$ *is any noise vector independent of* $\mathbf{h}$. **If we set** $b_i = -\sum_j a_{ij}p_j - \mathbf{W}_i^T\mathbb{E}_{\mathbf{e}}[\mathbf{e}] \ \forall i \in [m]$, **then** $\forall \ \delta \in (0,1)$,

$$\Pr\left(\frac{1}{m}\|\hat{\mathbf{h}} - \mathbf{h}\|_1 \leq \delta\right) \geq 1 - \sum_{i=1}^{m}\left((1-p_i)e^{-2\frac{(\delta' - \mathbf{W}_i^T(\mathbf{e} - \mathbb{E}_{\mathbf{e}}[\mathbf{e}]) + p_i a_{ii})^2}{\sum_{j=1, j \neq i}^{m} a_{ij}^2}}\right. \tag{40}$$

$$\left. + p_i e^{-2\frac{(\delta' - \mathbf{W}_i^T(\mathbf{e} - \mathbb{E}_{\mathbf{e}}[\mathbf{e}]) + (1-p_i) a_{ii})^2}{\sum_{j=1, j \neq i}^{m} a_{ij}^2}}\right) \tag{41}$$

*where* $a_{ij} = \mathbf{W}_i^T\mathbf{W}_j$, $\delta' = \ln(\frac{\delta}{1-\delta})$ *and* $\mathbf{W}_i$ *is the* $i^{th}$ *row of the matrix* $\mathbf{W}$ *cast as a column vector.*

*Proof.* Notice that,

$$\Pr(|\hat{h}_i - h_i| \geq \delta) = \Pr(|\hat{h}_i - h_i| \geq \delta \mid h_i = 0)\Pr(h_i = 0) \tag{42}$$

$$+ \Pr(|\hat{h}_i - h_i| \geq \delta \mid h_i = 1)\Pr(h_i = 1) \tag{43}$$

and from definition 1,

$$\hat{h}_i = \sigma\left(\sum_j a_{ij}h_j + b_i + \mathbf{W}_i^T\mathbf{e}\right) \tag{44}$$

Thus,

$$\Pr(|\hat{h}_i - h_i| \geq \delta) = (1-p_i)\Pr\left(\sigma\left(\sum_j a_{ij}h_j + b_i + \mathbf{W}_i^T\mathbf{e}\right) \geq \delta \mid h_i = 0\right)$$

$$+ p_i\Pr\left(\sigma\left(-\sum_j a_{ij}h_j - b_i - \mathbf{W}_i^T\mathbf{e}\right) \geq \delta \mid h_i = 1\right) \tag{45}$$

Notice that $\Pr(\sigma(\sum_j a_{ij}h_j + b_i + \mathbf{W}_i^T\mathbf{e}) \geq \delta \mid h_i = 0) = \Pr(\sum_j a_{ij}h_j + b_i + \mathbf{W}_i^T\mathbf{e} \geq \ln(\frac{\delta}{1-\delta}) \mid h_i = 0)$. **Let** $z_i = \sum_j a_{ij}h_j + b_i + \mathbf{W}_i^T\mathbf{e}$ and $\delta' = \ln(\frac{\delta}{1-\delta})$. Then, **setting** $b_i = -\mathbb{E}_{\mathbf{h}}[\sum_j a_{ij}h_j] - \mathbf{W}_i^T\mathbb{E}_{\mathbf{e}}[\mathbf{e}] = -\sum_j a_{ij}p_j$, using Chernoff's inequality on random variable $\mathbf{h}$, **for any** $t > 0$,

$$\Pr(z_i \geq \delta' \mid h_i = 0) \leq \frac{\mathbb{E}_{\mathbf{h}}[e^{tz_i}]}{e^{t\delta' - t\mathbf{W}_i^T(\mathbf{e} - \mathbb{E}_{\mathbf{e}}[\mathbf{e}])}} = \frac{\mathbb{E}_{\mathbf{h}}\left[e^{t\sum_{j \neq i} a_{ij}(h_j - p_j) - tp_i a_{ii}}\right]}{e^{t\delta' - t\mathbf{W}_i^T(\mathbf{e} - \mathbb{E}_{\mathbf{e}}[\mathbf{e}])}}$$

$$= \frac{\mathbb{E}_{\mathbf{h}}\left[\prod_{j \neq i} e^{ta_{ij}(h_j - p_j)}\right]}{e^{t(\delta' - t\mathbf{W}_i^T(\mathbf{e} - \mathbb{E}_{\mathbf{e}}[\mathbf{e}]) + p_i a_{ii})}} = \frac{\prod_{j \neq i}\mathbb{E}_{h_j}\left[e^{ta_{ij}(h_j - p_j)}\right]}{e^{t(\delta' - t\mathbf{W}_i^T(\mathbf{e} - \mathbb{E}_{\mathbf{e}}[\mathbf{e}]) + p_i a_{ii})}} \tag{46}$$

Setting $\bar{\delta} := \delta' - \mathbf{W}_i^T(\mathbf{e} - \mathbb{E}_{\mathbf{e}}[\mathbf{e}])$, we can rewrite the above inequality as

$$\Pr(z_i \geq \delta' \mid h_i = 0) \leq \frac{\prod_{j \neq i}\mathbb{E}_{h_j}\left[e^{ta_{ij}(h_j - p_j)}\right]}{e^{t(\bar{\delta} + p_i a_{ii})}} \tag{47}$$

Since the above inequality becomes identical to equation 25, the rest of the proof is similar to theorem 2. $\qquad\square$

**Theorem 2.** *Let each element of* $\mathbf{h} \in \mathbb{R}^m$ *follow BINS($\mathbf{p}, f_c, \mu_{\mathbf{h}}, \mathbf{l}_{\max}$) distribution and let* $\hat{\mathbf{h}}_{ReLU}(\mathbf{x}; \mathbf{W}, \mathbf{b})$ *be an auto-encoder signal recovery mechanism with Rectified Linear activation function (ReLU) and bias* $\mathbf{b}$ *for a measurement vector* $\mathbf{x} \in \mathbb{R}^n$ *such that* $\mathbf{x} = \mathbf{W}^T\mathbf{h}$. *If we set* $b_i \triangleq -\sum_j a_{ij}p_j\mu_{h_j} \ \forall i \in [m]$, **then** $\forall \ \delta \geq 0$,

$$\Pr\left(\frac{1}{m}\|\hat{\mathbf{h}} - \mathbf{h}\|_1 \leq \delta\right) \geq 1 - \sum_{i=1}^{m}\left(e^{-2\frac{(\delta + \sum_j (1-p_j)(l_{\max_j} - 2p_j\mu_{h_j})\max(0, a_{ij}))^2}{\sum_j a_{ij}^2 l_{\max_j}^2}}\right.$$

$$\left. + e^{-2\frac{(\delta + \sum_j (1-p_j)(l_{\max_j} - 2p_j\mu_{h_j})\max(0, -a_{ij}))^2}{\sum_j a_{ij}^2 l_{\max_j}^2}}\right) \tag{48}$$

*where $\mathbf{a}_i s$ are vectors such that*

$$a_{ij} = \begin{cases} \mathbf{W}_i^T \mathbf{W}_j & if \quad i \neq j \\ \mathbf{W}_i^T \mathbf{W}_i - 1 & if \quad i = j \end{cases} \tag{49}$$

$\mathbf{W}_i$ *is the $i^{th}$ row of the matrix $\mathbf{W}$ cast as a column vector.*

*Proof.* From definition 1 and the definition of $a_{ij}$ above,

$$\hat{h}_i = \max\{0, \sum_j a_{ij} h_j + h_i + b_i\}$$

$$\hat{h}_i - h_i = \max\{-h_i, \sum_j a_{ij} h_j + b_i\} \tag{50}$$

**Let** $z_i = \sum_j a_{ij} h_j + b_i$. Thus, $\hat{h}_i - h_i = \max\{-h_i, z_i\}$. Then, conditioning upon $z_i$,

$$\Pr(|\hat{h}_i - h_i| \leq \delta) = \Pr\left(|\hat{h}_i - h_i| \leq \delta \,\middle|\, h_i > 0, |z_i| \leq \delta\right) \Pr(|z_i| \leq \delta, h_i > 0) \tag{51}$$

$$+ \Pr\left(|\hat{h}_i - h_i| \leq \delta \,\middle|\, h_i > 0, |z_i| > \delta\right) \Pr(|z_i| > \delta, h_i > 0) \tag{52}$$

$$+ \Pr\left(|\hat{h}_i - h_i| \leq \delta \,\middle|\, h_i = 0, |z_i| \leq \delta\right) \Pr(|z_i| \leq \delta, h_i = 0) \tag{53}$$

$$+ \Pr\left(|\hat{h}_i - h_i| \leq \delta \,\middle|\, h_i = 0, |z_i| > \delta\right) \Pr(|z_i| > \delta, h_i = 0) \tag{54}$$

Since $\Pr\left(|\hat{h}_i - h_i| \leq \delta \,\middle|\, |z_i| \leq \delta\right) = 1$, we have,

$$\Pr(|\hat{h}_i - h_i| \leq \delta) \geq \Pr(|z_i| \leq \delta) \tag{55}$$

The above inequality is obtained by ignoring the positive terms that depend on the condition $|z_i| > \delta$ and marginalizing over $h_i$. **For any** $t > 0$, using Chernoff's inequality,

$$\Pr(z_i \geq \delta) \leq \frac{\mathbb{E}_{\mathbf{h}}\left[e^{tz_i}\right]}{e^{t\delta}} \tag{56}$$

**Setting** $b_i = -\sum_j a_{ij} \mu_j$, where $\mu_j = \mathbb{E}_{h_j}[h_j] = p_j \mu_{h_j}$,

$$\Pr(z_i \geq \delta) \leq \frac{\mathbb{E}_{\mathbf{h}}\left[e^{t \sum_j a_{ij}(h_j - \mu_j)}\right]}{e^{t\delta}} = \frac{\mathbb{E}_{\mathbf{h}}\left[\prod_j e^{ta_{ij}(h_j - \mu_j)}\right]}{e^{t\delta}} = \frac{\prod_j \mathbb{E}_{h_j}\left[e^{ta_{ij}(h_j - \mu_j)}\right]}{e^{t\delta}} \tag{57}$$

Let $T_j = \mathbb{E}_{h_j}\left[e^{ta_{ij}(h_j - \mu_j)}\right]$. Then,

$$T_j = (1 - p_j)e^{-ta_{ij}\mu_j} + p_j \mathbb{E}_{v \sim f_c(0^+, l_{\max}, \mu_h)}\left[e^{ta_{ij}(v - \mu_j)}\right] \tag{58}$$

where $f_c(a, b, \mu_h)$ denotes any arbitrary distribution in the interval $(a, b]$ with mean $\mu_h$. **If** $a_{ij} \geq 0$, let $\alpha = -\mu_j$ and $\beta = l_{\max_j} - \mu_j$ which essentially denote the lower and upper bound of $h_j - \mu_j$. Then,

$$T_j = (1 - p_j)e^{ta_{ij}\alpha} + p_j \mathbb{E}_{v \sim f_c(0^+, l_{\max_j}, \mu_{h_j})}\left[e^{ta_{ij}(v - \mu_j)}\right] \tag{59}$$

$$\leq (1 - p_j)e^{ta_{ij}\alpha} + p_j \mathbb{E}_v\left[\frac{\beta - (v - \mu_j)}{\beta - \alpha}e^{ta_{ij}\alpha} + \frac{(v - \mu_j) - \alpha}{\beta - \alpha}e^{ta_{ij}\beta}\right] \tag{60}$$

$$= (1 - p_j)e^{ta_{ij}\alpha} + \frac{\beta - (1 - p_j)\mu_{h_j}}{\beta - \alpha}p_j e^{ta_{ij}\alpha} + \frac{(1 - p_j)\mu_{h_j} - \alpha}{\beta - \alpha}p_j e^{ta_{ij}\beta} \tag{61}$$

$$= (1 - p_j)e^{ta_{ij}\alpha} + \frac{p_j \beta e^{ta_{ij}\alpha}}{\beta - \alpha} - \frac{p_j(1 - p_j)\mu_{h_j}}{(\beta - \alpha)}\left(e^{ta_{ij}\alpha} - e^{ta_{ij}\beta}\right) - \frac{p_j \alpha}{\beta - \alpha}e^{ta_{ij}\beta} \tag{62}$$

where the first inequality in the above equation is from the property of a convex function. **Define** $u = ta_{ij}(\beta - \alpha)$, $\gamma = -\frac{\alpha}{\beta - \alpha}$. Then,

$$T_j \leq e^{-u\gamma}\left[1 - p_j + \frac{p_j\beta}{\beta - \alpha} - \frac{p_j(1-p_j)\mu_{h_j}}{(\beta - \alpha)}(1 - e^u) - \frac{p_j\alpha}{\beta - \alpha}e^u\right] \tag{63}$$

$$= e^{-u\gamma}\left[1 + \frac{p_j\alpha}{\beta - \alpha} - \frac{p_j(1-p_j)\mu_{h_j}}{(\beta - \alpha)} - \left(\frac{p_j\alpha}{\beta - \alpha} - \frac{p_j(1-p_j)\mu_{h_j}}{(\beta - \alpha)}\right)e^u\right] \tag{64}$$

$$= e^{-u\gamma}\left[1 - \left(p_j\gamma + \frac{p_j(1-p_j)\mu_{h_j}}{(\beta - \alpha)}\right) + \left(p_j\gamma + \frac{p_j(1-p_j)\mu_{h_j}}{(\beta - \alpha)}\right)e^u\right] \tag{65}$$

$$\tag{66}$$

**Define** $\phi = p_j\gamma + \frac{p_j(1-p_j)\mu_{h_j}}{(\beta - \alpha)}$ and let $e^{g(u)} \triangleq T_j = e^{-u\gamma}(1 - \phi + \phi e^u)$. Then,

$$g(u) = -u\gamma + \ln(1 - \phi + \phi e^u) \implies g(0) = 0 \tag{67}$$

$$g'(u) = -\gamma + \frac{\phi e^u}{1 - \phi + \phi e^u} \implies g'(0) = -\gamma + \phi = -\gamma(1 - p) + \frac{p(1-p)\mu_h}{(\beta - \alpha)} \tag{68}$$

$$g''(u) = \frac{\phi(1 - \phi)e^u}{(1 - \phi + \phi e^u)^2} \tag{69}$$

$$g'''(u) = \frac{\phi(1 - \phi)(1 - \phi + \phi e^u)e^u(1 - \phi - \phi e^u)}{(1 - \phi + \phi e^u)^4} \tag{70}$$

Thus, for getting a maxima for $g''(u)$, we set $g'''(u) = 0$ which implies $1 - \phi - \phi e^u = 0$, or, $e^u = \frac{1 - \phi}{\phi}$. Substituting this $u$ in $g''(u) \leq 1/4$. By Taylor's theorem, $\exists c \in [0, u] \forall u > 0$ such that,

$$g(u) = g(0) + ug'(0) + \frac{u^2}{2}g''(c) \leq 0 - u\gamma(1 - p_j) + \frac{up_j(1-p_j)\mu_{h_j}}{(\beta - \alpha)} + u^2/8 \tag{71}$$

Thus we can upper bound $T_j$ as,

$$T_j \leq e^{u^2/8 - u\left(\gamma(1-p_j) - \frac{p_j(1-p_j)\mu_{h_j}}{(\beta - \alpha)}\right)} = e^{t^2 a_{ij}^2(\beta - \alpha)^2/8 + ta_{ij}(\beta - \alpha)\left(\frac{\alpha(1-p_j)}{\beta - \alpha} + \frac{p_j(1-p_j)\mu_{h_j}}{(\beta - \alpha)}\right)} \tag{72}$$

Substituting for $\alpha, \beta$, we get,

$$T_j \leq e^{t^2 a_{ij}^2 l_{\max_j}^2/8 + ta_{ij}(1-p_j)(-\mu_j + p_j\mu_{h_j})} = e^{\frac{t^2 a_{ij}^2 l_{\max_j}^2}{8}} \tag{73}$$

On the other hand, **if** $a_{ij} < 0$, then we can set $\alpha = \mu_j - l_{\max_j}$ and $\beta = \mu_j$ and proceeding similar to equation 59, we get,

$$T_j \leq e^{t^2 a_{ij}^2 l_{\max_j}^2/8 + t|a_{ij}|(1-p_j)(\mu_j - l_{\max_j} + p_j\mu_{h_j})} = e^{\frac{t^2 a_{ij}^2 l_{\max_j}^2}{8} - t|a_j|(1-p_j)(l_{\max_j} - 2p_j\mu_{h_j})} \tag{74}$$

Then, collectively, we can write $\Pr(z_i \geq \delta)$ as

$$\Pr(z_i \geq \delta) \leq \prod_j \frac{T_j}{e^{t\delta}} = e^{t^2 \sum_j a_{ij}^2 l_{\max_j}^2/8 - t\left(\delta + (1-p_j)(l_{\max_j} - 2p_j\mu_{h_j})\max(0, -a_{ij})\right)} \tag{75}$$

We similarly bound $\Pr(-z_i \geq \delta)$ by effectively flipping the sign of $a_{ij}$'s,

$$\Pr(-z_i \geq \delta) \leq e^{t^2 \sum_j a_{ij}^2 l_{\max_j}^2/8 - t\left(\delta + (1-p_j)(l_{\max_j} - 2p_j\mu_{h_j})\max(0, a_{ij})\right)} \tag{76}$$

Minimizing both 75 and 76 with respect to $t$ and applying union bound, we get,

$$\Pr(|\hat{h}_i - h_i| \geq \delta) \leq e^{-2\frac{(\delta + \sum_j(1-p_j)(l_{\max_j} - 2p_j\mu_{h_j})\max(0, a_{ij}))^2}{\sum_j a_{ij}^2 l_{\max_j}^2}} \tag{77}$$

$$+ e^{-2\frac{(\delta + \sum_j(1-p_j)(l_{\max_j} - 2p_j\mu_{h_j})\max(0, -a_{ij}))^2}{\sum_j a_{ij}^2 l_{\max_j}^2}} \quad \forall\, i \in [m] \tag{78}$$

Since the above bound holds for all $i \in [m]$, applying union bound on all the units yields the desired result.

$\square$

**Proposition 2.** *Let each element of $\mathbf{h}$ follow BINS($\mathbf{p}, f_c, \mu_{\mathbf{h}}, \mathbf{l}_{\max}$) distribution and let $\hat{\mathbf{h}} \in \mathbb{R}^m$ be an auto-encoder signal recovery mechanism with Rectified Linear activation function and bias $\mathbf{b}$ for a measurement vector $\mathbf{x} \in \mathbb{R}^n$ such that $\mathbf{x} = \mathbf{W}^T \mathbf{h} + \mathbf{e}$ where $\mathbf{e}$ is any noise random vector independent of $\mathbf{h}$. If we set $b_i \triangleq -\sum_j a_{ij} p_j \mu_{h_j} - \mathbf{W}_i^T \mathbb{E}_{\mathbf{e}}[\mathbf{e}] \; \forall i \in [m]$, then $\forall \, \delta \geq 0$,*

$$\Pr\left( \frac{1}{m} \|\hat{\mathbf{h}} - \mathbf{h}\|_1 \leq \delta \right) \geq 1 - \sum_{i=1}^m \left( e^{-2 \frac{(\delta - \mathbf{W}_i^T (\mathbf{e} - \mathbb{E}_{\mathbf{e}}[\mathbf{e}]) + \sum_j (1 - p_j)(l_{\max_j} - 2 p_j \mu_{h_j}) \max(0, a_{ij}))^2}{\sum_j a_{ij}^2 l_{\max_j}^2}} \right.$$
$$\left. + e^{-2 \frac{(\delta - \mathbf{W}_i^T (\mathbf{e} - \mathbb{E}_{\mathbf{e}}[\mathbf{e}]) + \sum_j (1 - p_j)(l_{\max_j} - 2 p_j \mu_{h_j}) \max(0, -a_{ij}))^2}{\sum_j a_{ij}^2 l_{\max_j}^2}} \right) \tag{79}$$

*where $\mathbf{a}_i$s are vectors such that*

$$a_{ij} = \begin{cases} \mathbf{W}_i^T \mathbf{W}_j & if \quad i \neq j \\ \mathbf{W}_i^T \mathbf{W}_i - 1 & if \quad i = j \end{cases} \tag{80}$$

$\mathbf{W}_i$ *is the $i^{th}$ row of the matrix $\mathbf{W}$ cast as a column vector.*

*Proof.* Recall that,

$$\hat{h}_i = \max\{0, \sum_j a_{ij} h_j + h_i + \mathbf{W}_i^T \mathbf{e} + b_i\} \tag{81}$$

$$\hat{h}_i - h_i = \max\{-h_i, \sum_j a_{ij} h_j + \mathbf{W}_i^T \mathbf{e} + b_i\} \tag{82}$$

**Let** $z_i = \sum_j a_{ij} h_j + b_i + \mathbf{W}_i^T \mathbf{e}$. Then, similar to theorem 2, conditioning upon $z_i$,

$$\Pr(|\hat{h}_i - h_i| \leq \delta) = \Pr\left( |\hat{h}_i - h_i| \leq \delta \,\Big|\, h_i > 0, |z_i| \leq \delta \right) \Pr(|z_i| \leq \delta, h_i > 0) \tag{83}$$

$$+ \Pr\left( |\hat{h}_i - h_i| \leq \delta \,\Big|\, h_i > 0, |z_i| > \delta \right) \Pr(|z_i| > \delta, h_i > 0) \tag{84}$$

$$+ \Pr\left( |\hat{h}_i - h_i| \leq \delta \,\Big|\, h_i = 0, |z_i| \leq \delta \right) \Pr(|z_i| \leq \delta, h_i = 0) \tag{85}$$

$$+ \Pr\left( |\hat{h}_i - h_i| \leq \delta \,\Big|\, h_i = 0, |z_i| > \delta \right) \Pr(|z_i| > \delta, h_i = 0) \tag{86}$$

Since $\Pr\left( |\hat{h}_i - h_i| \leq \delta \,\Big|\, |z_i| \leq \delta \right) = 1$, we have,

$$\Pr(|\hat{h}_i - h_i| \leq \delta) \geq \Pr(|z_i| \leq \delta) \tag{87}$$

**For any** $t > 0$, using Chernoff's inequality for the random variable $\mathbf{h}$,

$$\Pr(z_i \geq \delta) \leq \frac{\mathbb{E}_{\mathbf{h}}[e^{t z_i}]}{e^{t \delta}} \tag{88}$$

**Setting** $b_i = -\sum_j a_{ij} \mu_j - \mathbf{W}_i^T \mathbb{E}_{\mathbf{e}}[\mathbf{e}]$, where $\mu_j = \mathbb{E}_{h_j}[h_j] = p_j \mu_{h_j}$,

$$\Pr(z_i \geq \delta) \leq \frac{\mathbb{E}_{\mathbf{h}}\left[ e^{t \sum_j a_{ij}(h_j - \mu_j)} \right]}{e^{t\delta - t \mathbf{W}_i^T (\mathbf{e} - \mathbb{E}_{\mathbf{e}}[\mathbf{e}])}} = \frac{\mathbb{E}_{\mathbf{h}}\left[ \prod_j e^{t a_{ij}(h_j - \mu_j)} \right]}{e^{t\delta - t \mathbf{W}_i^T (\mathbf{e} - \mathbb{E}_{\mathbf{e}}[\mathbf{e}])}} = \frac{\prod_j \mathbb{E}_{h_j}\left[ e^{t a_{ij}(h_j - \mu_j)} \right]}{e^{t\delta - t \mathbf{W}_i^T (\mathbf{e} - \mathbb{E}_{\mathbf{e}}[\mathbf{e}])}} \tag{89}$$

Setting $\bar{\delta} := \delta - \mathbf{W}_i^T (\mathbf{e} - \mathbb{E}_{\mathbf{e}}[\mathbf{e}])$, we can rewrite the above inequality as

$$\Pr(z_i \geq \delta) \leq \frac{\prod_j \mathbb{E}_{h_j}\left[ e^{t a_{ij}(h_j - \mu_j)} \right]}{e^{t\bar{\delta}}} \tag{90}$$

Since the above inequality becomes identical to equation 57, the rest of the proof is similar to theorem 2. $\qquad \square$

**Theorem 3.** *(Uncorrelated Distribution Bound): If data is generated as* $\mathbf{x} = \mathbf{W}^T\mathbf{h}$ *where* $\mathbf{h} \in \mathbb{R}^m$ *has covariance matrix* $\mathrm{diag}(\zeta)$, $(\zeta \in \mathbb{R}^{+^m})$ *and* $\mathbf{W} \in \mathbb{R}^{m \times n}$ $(m > n)$ *is such that each row of* $\mathbf{W}$ *has unit length and the rows of* $\mathbf{W}$ *are maximally incoherent, then the covariance matrix of the generated data is approximately spherical (uncorrelated) satisfying,*

$$\min_{\alpha}\|\mathbf{\Sigma} - \alpha\mathbf{I}\|_F \le \sqrt{\frac{1}{n}\left(m\|\zeta\|_2^2 - \|\zeta\|_1^2\right)} \tag{91}$$

*where* $\mathbf{\Sigma} = \mathbb{E}_{\mathbf{x}}[(\mathbf{x} - \mathbb{E}_{\mathbf{x}}[\mathbf{x}])(\mathbf{x} - \mathbb{E}_{\mathbf{x}}[\mathbf{x}])^T]$ *is the covariance matrix of the generated data.*

*Proof.* Notice that,

$$\mathbb{E}_{\mathbf{x}}[\mathbf{x}] = \mathbf{W}^T\mathbb{E}_{\mathbf{h}}[\mathbf{h}] \tag{92}$$

Thus,

$$\mathbb{E}_{\mathbf{x}}[(\mathbf{x} - \mathbb{E}_{\mathbf{x}}[\mathbf{x}])(\mathbf{x} - \mathbb{E}_{\mathbf{x}}[\mathbf{x}])^T] = \mathbb{E}_{\mathbf{h}}[(\mathbf{W}^T\mathbf{h} - \mathbf{W}^T\mathbb{E}_{\mathbf{h}}[\mathbf{h}])(\mathbf{W}^T\mathbf{h} - \mathbf{W}^T\mathbb{E}_{\mathbf{h}}[\mathbf{h}])^T] \tag{93}$$

$$= \mathbb{E}_{\mathbf{h}}[\mathbf{W}^T(\mathbf{h} - \mathbb{E}_{\mathbf{h}}[\mathbf{h}])(\mathbf{h} - \mathbb{E}_{\mathbf{h}}[\mathbf{h}])^T\mathbf{W}] \tag{94}$$

$$= \mathbf{W}^T\mathbb{E}_{\mathbf{h}}[(\mathbf{h} - \mathbb{E}_{\mathbf{h}}[\mathbf{h}])(\mathbf{h} - \mathbb{E}_{\mathbf{h}}[\mathbf{h}])^T]\mathbf{W} \tag{95}$$

Substituting the covariance of $\mathbf{h}$ as $\mathrm{diag}(\zeta)$,

$$\Sigma = \mathbb{E}_{\mathbf{x}}[(\mathbf{x} - \mathbb{E}_{\mathbf{x}}[\mathbf{x}])(\mathbf{x} - \mathbb{E}_{\mathbf{x}}[\mathbf{x}])^T] = \mathbf{W}^T\,\mathrm{diag}(\zeta)\mathbf{W} \tag{96}$$

Thus,

$$\|\mathbf{\Sigma} - \alpha\mathbf{I}\|_F^2 = \mathrm{tr}\left((\mathbf{W}^T\,\mathrm{diag}(\zeta)\mathbf{W} - \alpha\mathbf{I})(\mathbf{W}^T\,\mathrm{diag}(\zeta)\mathbf{W} - \alpha\mathbf{I})^T\right) \tag{97}$$

$$= \mathrm{tr}\left(\mathbf{W}^T\,\mathrm{diag}(\zeta)\mathbf{W}\mathbf{W}^T\,\mathrm{diag}(\zeta)\mathbf{W} + \alpha^2\mathbf{I} - 2\alpha\mathbf{W}^T\,\mathrm{diag}(\zeta)\mathbf{W}\right) \tag{98}$$

Using the cyclic property of trace,

$$\|\mathbf{\Sigma} - \alpha\mathbf{I}\|_F^2 = \mathrm{tr}\left(\mathbf{W}\mathbf{W}^T\,\mathrm{diag}(\zeta)\mathbf{W}\mathbf{W}^T\,\mathrm{diag}(\zeta) + \alpha^2\mathbf{I} - 2\alpha\mathbf{W}\mathbf{W}^T\,\mathrm{diag}(\zeta)\right) \tag{99}$$

$$= \|\mathbf{W}\mathbf{W}^T\,\mathrm{diag}(\zeta)\|_F^2 + \alpha^2 n - 2\alpha\sum_{i=1}^{m}\zeta_i \tag{100}$$

$$\le \left(\sum_{i=1}^{m}\zeta_i^2\right)(1 + \mu^2(m-1)) + \alpha^2 n - 2\alpha\sum_{i=1}^{m}\zeta_i \tag{101}$$

Finally minimizing w.r.t $\alpha$, we get $\alpha^* = \frac{1}{n}\sum_{i=1}^{m}\zeta_i$. Substituting this into the above inequality, we get,

$$\min_{\alpha}\|\mathbf{\Sigma} - \alpha\mathbf{I}\|_F^2 \le \left(\sum_{i=1}^{m}\zeta_i^2\right)(1 + \mu^2(m-1)) + \frac{1}{n}\left(\sum_{i=1}^{m}\zeta_i\right)^2 - \frac{2}{n}\left(\sum_{i=1}^{m}\zeta_i\right)^2 \tag{102}$$

$$= \left(\sum_{i=1}^{m}\zeta_i^2\right)(1 + \mu^2(m-1)) - \frac{1}{n}\left(\sum_{i=1}^{m}\zeta_i\right)^2 \tag{103}$$

$$\tag{104}$$

Since the weight matrix is maximally incoherent, using Welch bound, we have that, $\mu \in \left[\sqrt{\frac{m-n}{n(m-1)}}, 1\right]$. Plugging the lower bound of $\mu$ (maximal incoherence) for any fixed $m$ and $n$ into the above bound yields,

$$\min_{\alpha}\|\mathbf{\Sigma} - \alpha\mathbf{I}\|_F^2 \le \left(\sum_{i=1}^{m}\zeta_i^2\right)\left(1 + \frac{m-n}{n(m-1)}(m-1)\right) - \frac{1}{n}\left(\sum_{i=1}^{m}\zeta_i\right)^2 \tag{105}$$

$$= \left(\sum_{i=1}^{m}\zeta_i^2\right)\left(1 + \frac{m-n}{n}\right) - \frac{1}{n}\left(\sum_{i=1}^{m}\zeta_i\right)^2 \tag{106}$$

$$= \frac{1}{n}\left(m\|\zeta\|_2^2 - \|\zeta\|_1^2\right) \tag{107}$$

$\square$

## 2 SUPPLEMENTARY EXPERIMENTS

### 2.1 SUPPLEMENTARY EXPERIMENTS FOR SECTION 5.1

Here we show the recovery error (APRE) for signals generated with coherent weight matrix, and as expected the recovery result is poor and the values of $c$ and $\Delta b$ are unpredictable. The minimum average percentage recovery error we got for continous signal is 45.75, and for binary signal is 32.63.

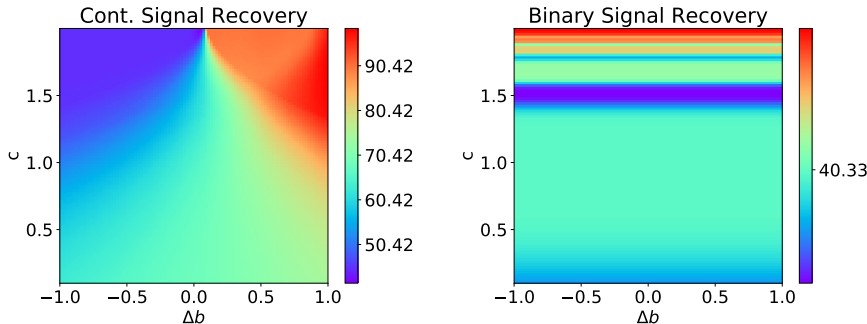

Figure 5: Error heatmap showing optimal values of $c$ and $\Delta b$ for recovering continous (left) and binary (right) signal using coherent weights.

### 2.2 SUPPLEMENTARY EXPERIMENTS FOR SECTION 5.2

Fig. 6 shows that the coherence of orthogonal initialized weight matrix is more incoherent as compared to the ones that using Gaussian based initialization.

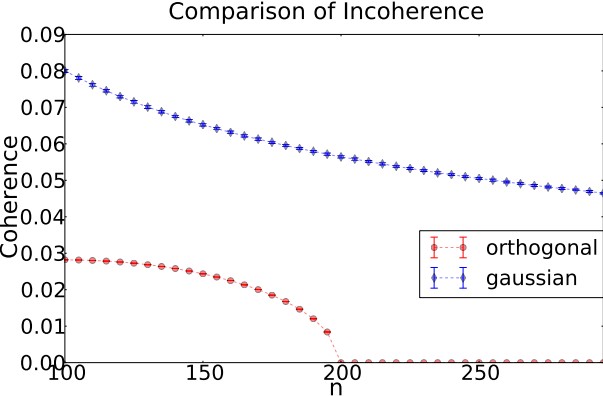

Figure 6: Coherence of orthogonal and Gaussian weight matrix with varying dimensions.

### 2.3 SUPPLEMENTARY EXPERIMENTS FOR SECTION 5.3

For noisy signal recovery we add independent Gaussian noise to data with mean 100 and standard deviation ranging from 0.01 to 0.2. Note that the data is normally within the range of $[-1, 1]$, so the noise is quite significant when we have a standard deviation $> 0.1$. It is clear that even in noisy case AE can recover the dictionary (see fig. 7). However, the recovery is not very strong when the noise is large $> 0.1$ for continous signals, which is because 1) the precise value in this case is continous and thus is more influenced by the noise, 2) the dictionary recovery is poor, which result poor signal recovery. On the other hand, the recovery is robust in case of recovering binary signals. Similar results were found on the APRE of recovered hidden signals. The reason for more robust recovery for binary signal is that 1) the information content is lower and 2) we binarize the recovered hidden signal by thresholding it, which further denoised the recovery. When optimizing the AE objective for binary

singal recovery case, we did a small trick to simulate the binarization of the signal. From our analysis (see Theorem 1), a recovery error $\delta = 0.5$ is reasonable as we can binarize the recovery using some threshold. However, when optimizing the AE using gradient based method we are unable to do this. To simulate this effect, we offset the pre-activation by a constant $k$ and multiply the pre-activation by a constant $c$, so that it signifies the input and push the post-activation values towards $0$ and $1$. In other words, we optimize the following objective when doing binary signal recovery:

$$\hat{\mathbf{W}} = \arg \min_{\mathbf{W}} \mathbb{E}_{\mathbf{x}} \left[ \| \mathbf{x} - \mathbf{W}^T \sigma \left( c \left\{ \mathbf{W}(\mathbf{x} - \mathbb{E}_{\mathbf{x}}[\mathbf{x}]) + k \right\} \right) \|^2 \right], \qquad \text{where} \| \mathbf{W_i} \|_2^2 = 1 \forall i \quad (108)$$

where $\sigma$ is the sigmoid function. We find set $c = 6$ and $k = -0.6$ is sufficient to saturate the sigmoid and simulate the binarization of hidden signals.

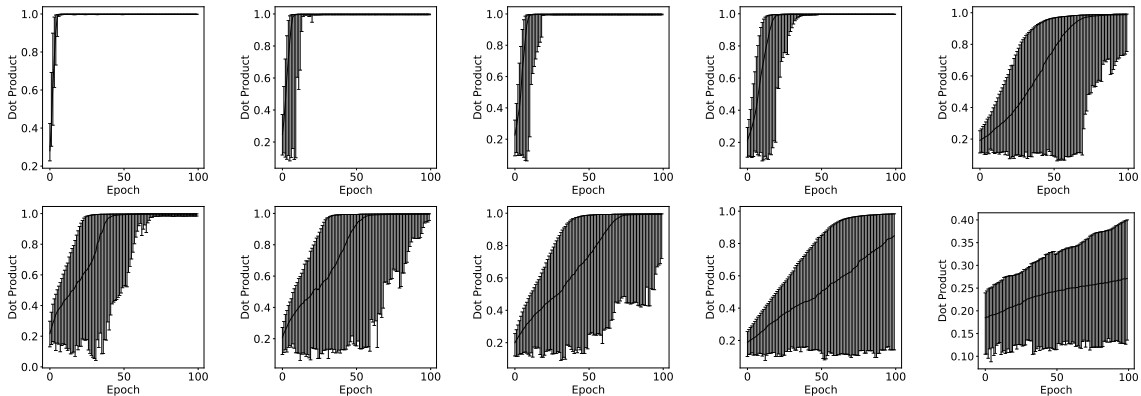

Figure 7: Cosine similarity between greedy paired rows of $\mathbf{W}$ and $\hat{\mathbf{W}}$ for noisy binary (upper) and continous (lower) recovery. From left to right the noise stand deviations are 0.01, 0.02, 0.05, 0.1, 0.2, respectively. The upper, mid and lower bar represent the 95th, 50th and 5th percentile.

Table 1: Average percentage recovery error for noisy AE recovery.

| Noise std. | 0.01 | 0.02 | 0.05 | 0.1 | 0.2 |
|---|---|---|---|---|---|
| Continous APRE | 2.06 | 1.63 | 9.48 | 34.16 | 56.79 |
| Binary APRE | 0.15 | 0.16 | 0.18 | 1.56 | 4.00 |

