# OpenReview forum: "On Optimality Conditions for Auto-Encoder Signal Recovery"
_ICLR.cc/2018/Conference — Reject_

### Official Review · AnonReviewer2 · 2017-11-22
**Recovery guarantees for auto-encoders; a rather unclear paper, with insufficient comparisons to previous work, and no algorithmic/practical perspectives.**

**Rating:** 4
**Confidence:** 3

**Review:**

*Summary*
The paper studies recovery guarantees within the context of auto-encoders. Assuming a noise-corrupted linear model for the inputs x's, the paper looks at some sufficient properties (e.g., over the generating dictionary denoted by W) to recover the true underlying sparse signals (denoted by h). Several settings of increasing complexity are considered (from binary signals with no noise to noisy continuous signals). Evaluations are carried out on synthetic examples to highlight the theoretical findings.

The paper is overall difficult to read. Moreover, and importantly, no algorithmic perspectives are presented in the paper, in the sense that we do not know whether practical procedures would lead to W's satisfying the appropriate properties (unlike (not-mentioned) recent results for dictionary learning/ICA; see detailed comments). Also, assumptions are made (e.g., knowledge about expectations of h and x) for which it is unclear to see how practical/limiting they are. Finally (and as further discussed below), the paper does not sufficiently discuss related work.

(note: I have not reviewed the appendix and supplementary material)

*Detailed comments*

-I think there is an insufficient literature review about recent recovery results in the context of sparse coding, dictionary learning and ICA (see some references at the bottom of the review). I think this is all the more important as the paper tries to draw connections with ICA (see Sec. 4.4).
Given that the paper positions itself on a theoretical level, detailed comparisons with existing sample complexities obtained in previous work for related models (e.g., sparse coding, dictionary learning and ICA) must be provided.

-To the best of my understanding of the paper, the guarantees are about h_hat and the true h. It therefore seems that the paper's approach is very close to standard sparse inverse problems, up to the difference due to the (non-identity) activation function. If this is indeed the case, the paper should discuss its results when the activation is identity to see whether known results are recovered.

-"...we consider linear activation s_d because it is a more general case.": Just after this statement, it is mentioned that non-linear activations are used in practice. Could this statement be therefore clarified?

-Sec. 2 is unclear. For instance, it is not easy to see how one go from (1) to (2). Moreover, the concept of "AE framework" is not well defined.

-In the bottom of page 3, why are p_i and (1-p_i) discarded?

-In practice, how can we set the appropriate value of b_i?

-What is the practical sense of being able to have access to E_h[x], E_x[x], and E_h[h]?

-In Proposition 1 and 2, if the noise e is indeed random, it means the right-hand sides are also random variables. Then, what does the probability statement Pr mean on the left-hand side? Is is conditioned on the draw of e? Some clarifications are required.

-Typo page 7: "...that used to generate the data." --> "... used to generate the data."
-Typo page 9: "...data are then generate..." --> "...data are then generated..."

-In Sec. 5.3, to match W_hat and W, the Hungarian algorithm can probably be used.

*References*

(Arora2012) Arora, S.; Ge, R.; Moitra, A. & Sachdeva, S. Provable ICA with unknown Gaussian noise, with implications for Gaussian mixtures and autoencoders Advances in Neural Information Processing Systems (NIPS), 2012, 2375-2383

(Arora2013) Arora, S.; Ge, R. & Moitra, A. New algorithms for learning incoherent and overcomplete dictionaries preprint arXiv:1308.6273, 2013

(Chatterji2017) Chatterji, N. S. & Bartlett, P. L. Alternating minimization for dictionary learning with random initialization preprint arXiv:1711.03634, 2017

(Gribonval2015) Gribonval, R.; Jenatton, R. & Bach, F. Sparse and spurious: dictionary learning with noise and outliers IEEE Transactions on Information Theory, 2015, 61, 6298-6319

(Sun2015) Sun, J.; Qu, Q. & Wright, J. Complete dictionary recovery over the sphere Sampling Theory and Applications (SampTA), 2015 International Conference on, 2015, 407-410

---

> ### Author Response · Authors · 2018-01-05
> **Response to reviewer 2**
>
> Thank you for your comments.
>
> We would like to stress that our goal in this paper was to study the signal  recovery properties of auto-encoders rather than studying them from a dictionary learning point of view which is a separate problem on its own.
>
> Regarding not providing algorithmic perspective.
> We are interested in properties that would lead to optimal solutions using auto-encoders from a signal recovery perspective, and thus better understand auto-encoders. There is no algorithmic perspective since 1) when the dictionary is known, we can get the solution analytically, 2) in case the dictionary is unknown we solve it using gradient descent. In the second case, we do not have guarantees for the recovery of the hidden signals, however, we have shown that empirically the recovery is strong using the theories that we developed.
>
> Regarding the linearity of the activation function s_d, we apologize for the confusion, we have reworded to make it more clear. s_d is the decoding activation function, linear activation is more general as it covers a wider numerical range. The latter sentence meant to say 1) the activations for both s_d and s_e can be non-linear in practice; 2) since linear s_d can handle the case for non-linear s_d, the previous statement is still true.
>
> On page 3, p_i and 1-p_i are constants with respect to the data and therefore it is suffice to analyse the other terms.
>
> Regarding how to set the value of b_i, as stated in theorem 1 and 2, b_i can be set analytically based on weights and p_i. In practice, since we do not know the sparsity level, we can set it in two ways, 1) treat p_i as hyper parameters, 2) treat p_i as parameters of the model.
>
> E_h[x] and E_x[x] are all data mean, and E_h[h] are mean of the hidden activations.
>
> We have fixed other minor problems mentioned in your reviews.

---

### Official Review · AnonReviewer3 · 2017-11-26
**On optimality conditions for auto-encoder signal recovery**

**Rating:** 5
**Confidence:** 4

**Review:**

This papers proposes to analyze auto-encoders under sparsity constraints of an underlying signal to be recovered.
Based on concentration inequality, the reconstruction provided for a simple class of functions is guaranteed to be accurate in l1 norm with high probability.
The proof techniques are classical, but the results seem novel as far as I know.
As an open question, could the results be given for other lp norms, in particular for infinity-norm? Indeed, this is a privileged norm for support recovery.



Presentation issues:
- section should be Section when stating for instance "section 1". Idem for eq, equation, assumption...
- bold fonts for vectors are randomly used: some care should be given to harmonizing symbols fonts.
- equations should be cited with brackets

References issues:
- harmonize citations: if you add first name for some authors add it for all references: why writing Roland Makhzani and J. Wright?

- Candes -> Cand\`es

- Consider citing "Sparse approximate solutions to linear systems", Natarajan 1995 when mentioning Amaldi and Kann 1998.



Specific comments:
page 1:
- hasn't -> has not.

page 2:
- "activation function": at this stage s_e and s_d are just functions. What is the "activation" refers to? Also a clarification on the space they act on should be stated. Idem for b_e and b_d.
- "the identity of h in eq. 1 is only well defined in the presence of l1 regularization due to the over-completeness of the dictionary" : this is implicitly stating the uniqueness of the Lasso. Not that it is well known that there are cases where the Lasso is non-unique. Please, clarify your statement accordingly.
- for simplicity b_d could be removed here.
- in (4) it would be more natural to write f_j(h_j) instead of f(h_j)
- "has is that to be bounded"-> is boundedness?
- what is l_max_j here? moreover the bold letters seem to represent vectors but this should be state explicitly somewhere.

page 3:
- what is the precise meaning of "distributed" when referring to representation
- In remark 1: the font has changed weirdly for W and h.
- "two class"->two classes
- Definition 1: again what is a precise definition of activation function?
- "if we set": bold issue.
- b should b_e in Theorem 1, right? Also, please recall the definition of the sigmoid function here. Moreover l_max and mu_h seem useless in this theorem... why referring to them?
- "if the rows of the weight matrix is"-> if the rows of the weight matrix are

page 4:
- Proposition 1 could be stated as a theorem and Th.1 as a corollary (with e=0). The same is true for proposition 2 I suspect.
- Again the influence of l_max and mu_h are none here...
- Please, provide the definition of the ReLu function here. Is this just x->x_+ ?

page 6:
- R^+m -> font issue again.
- "are maximally incoherent": what is the precise meaning of this statement?
- what the motivation for Theorem 3? This should be discussed.
- De-noising -> de-noising
- the discussion after (15) should be made more precise.

page 7:
- Figure 1 and 2 should be postponed to page 8.
- in Eq. (16) one needs to known E_h(x) and E_h_i(h_i), but I suspect this quantity are usually unknown to the practitioner. Can the author comment on that?

page 8:
- "the recovery is denoised through thresholding": where is this step analyzed?

page 9:
- figure 3: sparseness-> sparsity; also what is the activation function used here?
- "are then generate"->are then generated
- "by greedily select"->by greedily selecting
- "the the"
- "and thus pre-process"-> and thus pre-processing


Supplementary:
page 1:
- please define \sigma, and its simple properties used along the proof.

page 2:
- g should be g_j (in eq 27 - > 31)
- overall this proof relies on ingredients such as the one used for Hoeffding's inequality.
Most ingredients could be taken from standard tools on concentration (see for instance Boucheron, Lugosi, Massart: "Concentration Inequalities: A Nonasymptotic Theory of Independence", 2013).
Moreover, some elements should be factorized as they are shared among the next proofs. This should reduce the size of the supplementary dramatically.

page 7:
- Eq. (99): it should be reminded that W_ii=1 here.
- the upper bound used on \mu to get equation 105 seems to be in the wrong order.

---

> ### Author Response · Authors · 2018-01-05
> **Response to reviewer 3**
>
> Thank you for your diligent reviews and detailed comments.
>
> Our results can be extended trivially to other norms using the standard norm equalities; however a non-trivial bound for specific norms (Eg. infinity norm) may need additional work.
>
> We have harmonized the citations as suggested by the reviewer.  Following are clarifications to some of the questions.
>
> What is the "activation" refers to?
> We use the term activation function in accordance to the terminology used commonly for auto-encoders.
>
> There are cases where the Lasso is non-unique.
> Indeed there are cases when the solution to LASSO is not unique (Eg. when the line y=Dw aligns with |w|=c for some c). We were referring to the general case when the solution to y=Dw is not unique at all when D is over-complete. In these cases some sort of constraint like L1 or L2 penalty is needed to make the solution unique.
>
> in (4) it would be more natural to write f_j(h_j) instead of f(h_j)
> We use the notation f(h_j) instead of f_j(h_j) to stress that all units are identical.
>
>  what is the precise meaning of "distributed" when referring to representation
> Distributed representation is a terminology used in deep learning which implies multiple hidden units participate together to represent one sample instead of a single hidden unit corresponding to a single sample. It is an efficient way of encoding patterns.
>
> l_max and mu_h seem useless in this theorem
> I_max and mu_h is required for the data generating distribution BINS defined in our paper; they are the sufficient statistics.
>
> ReLu
> ReLU is x -> max(0, x)
>
> "are maximally incoherent": what is the precise meaning of this statement?
> The vectors in a matrix are maximally incoherent if the minimum of the angle between every pair of vectors is maximized. This angle is given by the Welch bound.
>
> what the motivation for Theorem 3?
> As mentioned at the beginning of this subsection, the motivation behind theorem 3 is to gain some insights for the generated data.
>
> Quantities of  E_h(x) and E_h_i(h_i)
> Yes the quantities E_h(x) and E_h_i(h_i) are unknown, but notice this value is equal to W E_x[x] where x is observed. So as long as W can be recovered, the quantity E_h[h] can be computed.
>
> Regarding "the recovery is denoised through thresholding", we didn't analyze it. We mention this based on the intuition that the signal is recovered with epsilon error with high probability. So if the signal magnitude is large enough compared to noise, a simple thresholding should work in separating signal from noise.
>
> In figure 3, ReLU is used for continuous recovery and sigmoid is used for binary case, as mentioned in theorem 1 in section 4.1 and theorem 2 in section 4.2

---

### Official Review · AnonReviewer1 · 2017-11-28
**Recovering sparse signals via ReLU/Sigmoid functions**

**Rating:** 5
**Confidence:** 4

**Review:**

This paper considers the following model of a signal x = W^T h + b, where h is an m-dimensional random sparse vector, W is an m by n matrix, b is an n dimensional fixed bias vector. The random vector h follows an iid sparse signal model, each coordinate independently have some probability of being zero, and the remaining probability is distributed  among nonzero values according to some reasonable pdf/pmf. The task is to recover h, from the observation x via the activation functions like Sigmoid or ReLU. For example, \hat{h} = Sigmoid(W^T h + b).

The authors then show that, under the random sparsity model of h, it is possible to upper bound the probability P(||h-\hat{h}|| > \delta. m) in terms of the parameters of the distribution of h and W and b. In some cases noise can also be tolerated. In particular, if W is incoherent (columns being near-orthonormal), then the guarantee is stronger. As far as I understood, the proofs make sense - they basically use Chernoff-bound type argument.

It is my impression that a lot of conditions have to be satisfied for the recovery guarantee to be meaningful. I am unsure if real datasets will satisfied so many conditions. Also, the usual objective of autoencoders is to denoise  - i.e. recover x, without any access to W. The authors approach in this vein seem to be only empirical. Some recent works on associative memory also assume the sparse recovery model - connections to this literature would have been of interest. It is also not clear why compressed sensing-type recovery using a single ReLU or Sigmoid would be of interest: are their complexity benefits?

---

> ### Author Response · Authors · 2018-01-05
> **Response to reviewer 1**
>
> Thanks for your comments.
>
> Regarding the questions. All real datasets of course do not exactly satisfy the assumptions made in the paper. It also depends on how we model the data. For instance, modeling images at patch levels would be conducive to the assumptions made for real images while modeling them at the image level itself may not. To further defend that the assumptions would hold at patch level, consider that independent component analysis (ICA) is commonly used to model real images at patch level. As discussed in the paper, ICA is a special case of our model where the signal is further assumed to be sparse for recovery using the mechanism discussed in the paper. It is well known that the sparsity assumption usually holds in practice.
>
> Regarding the use of ReLU and Sigmoid, other non-linearities can be used for signal recovery as well as long as they satisfy certain criteria.

---

### Decision · Program_Chairs · 2018-01-29
**ICLR 2018 Conference Acceptance Decision**

**Decision:**

Reject

**Comment:**

- The paper is overall difficult to read and would benefit from a revised presentation.
- The practical relevance of the recovery conditions and algorithmic consequences of the work is not sufficiently clear or  convincing.